# Just Adjust One Prompt: Enhancing In-Context Dialogue Scoring via Constructing the Optimal Subgraph of Demonstrations and Prompts

**Jiashu Pu**[1], **Ling Cheng**[3], **Lu Fan**[4], **Tangjie Lv**[1] **and Rongsheng Zhang**[1,2,†]

[1] Fuxi AI Lab, NetEase Inc., Hangzhou, China
[2] Zhejiang University, Hangzhou, China
[3] Singapore Management University, Singapore
[4] The Hong Kong Polytechnic University, Kowloon, Hong Kong, China
{pujiashu,hzlvtangjie,zhangrongsheng}@corp.netease.com
lingcheng.2020@phdcs.smu.edu.sg; cslfan@comp.polyu.edu.hk

## Abstract

The use of modern Large Language Models (LLMs) as chatbots still has some problems such as hallucinations and lack of empathy. Identifying these issues can help improve chatbot performance. The community has been continually iterating on reference-free dialogue evaluation methods based on large language models (LLMs) that can be readily applied. However, many of these LLM-based metrics require selecting specific datasets and developing specialized training tasks for different evaluation dimensions (e.g., coherence, informative). The developing step can be time-consuming and may need to be repeated for new evaluation dimensions. To enable efficient and flexible adaptation to diverse needs of dialogue evaluation, we propose a dimension-agnostic scoring method that leverages the in-context learning (ICL) capability of LLMs to learn from human scoring to the fullest extent. Our method has three key features. To begin with, rather than manual prompt crafting, we propose automatically generating prompts, allowing the LLM to observe human labels and summarize the most suitable prompt. Additionally, since the LLM has a token limit and ICL is sensitive to demonstration variations, we train a selector to finely customize demonstrations and prompts for each dialogue input. Finally, during inference, we propose to request the LLM multiple times with a subgraph of demonstrations and prompts that are diverse and suitable to maximize ICL from various human scoring. We validate the efficacy of our method on five datasets, even with a small amount of annotated data, our method outperforms all strong baselines. Code is available at EMNLP2023-ADOROR.

## 1 Introduction

Although modern language models have made significant strides in language fluency, informativeness, and user understanding (Taecharungroj,

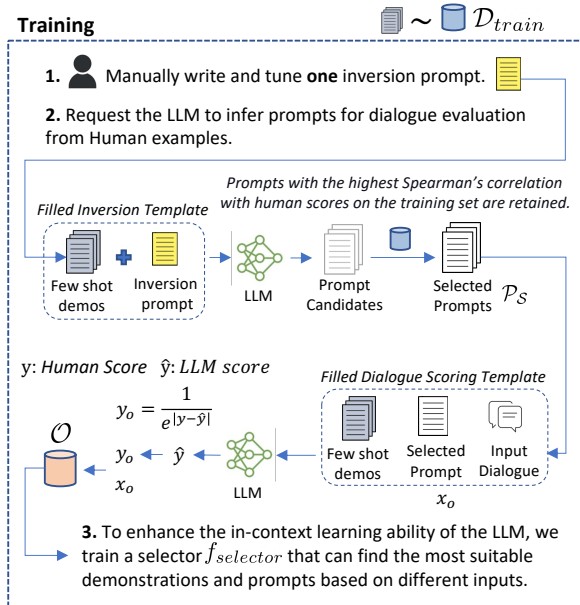

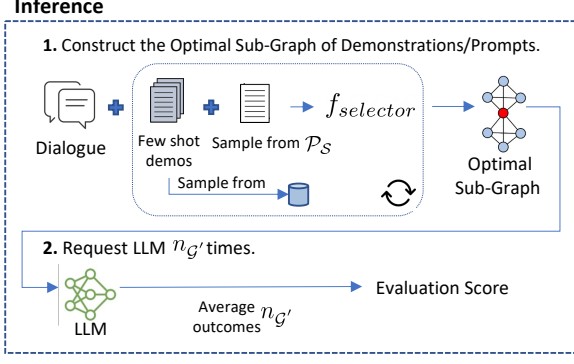

Figure 1: Overview of our ADOROR method.

2023; Yang et al., 2023), even the most advanced LLMs (such as ChatGPT) still exhibit notable flaws, including hallucination (Alkaissi and McFarlane, 2023; Ji et al., 2023) and a lack of emotion perception when used as a chatbot (Tlili et al., 2023). By precisely evaluating the dialogue, we can identify areas for improvement and make necessary adjustments to enhance the chatbot's performance.

The gold standard for dialogue evaluation is human scoring (Adewumi et al., 2022), but human

---

[†] Corresponding Author.

scoring is subjective (Zhang et al., 2020), time-consuming and expensive (Pang et al., 2020; Yeh et al., 2021; Mehri and Eskenazi, 2020b). In recent years, many automated reference-free evaluation schemes have emerged; these schemes infuse pre-trained LLMs with prior knowledge at training time by selecting data and designing specific tasks for various evaluation dimensions (Huang et al., 2020; Sinha et al., 2020; Zhong et al., 2022; Zhang et al., 2022). However, the dimension of interest may change with the evolution of LLMs, for modern LLMs, it is more valuable for us to focus on persona consistency, the presence of bias (Ray, 2023), etc. Whenever a new dimension is to be evaluated, previous approaches necessitate a two-stage procedure of data selection and training task formulation, which is inefficient. Not only that, they only support contexts and responses as input, and cannot imbue specific background information, which can pose a challenge when dealing with diverse scoring criteria. As an illustration, if we utilize a chatbot as an academic aide (Tlili et al., 2023), it is imperative that the chatbot customizes its replies to the various age groups of the students. Responses to younger students ought to be engaging and captivating, whereas responses to older students should prioritize a more formal and proficient tone. Apart from this, in specific use cases, such as those involving medical chatbots, the standards for assessing responses must be custom-designed (Chow et al., 2023). Our view is that it is unfeasible to prepare an off-the-shelf pre-train-based metric that can adequately accommodate all scoring criteria.

To enable efficient adaptation to novel evaluation dimensions or unique scenarios, we design a new dialogue evaluation method — Enhancing In-Context Dialogue Scoring through constructing the Optimal Sub-Graph Of Demonstrations and Prompts (ADOROR), with an overview in Figure 1. With in-context learning (ICL) at its core, our approach requires only a few labeled data and has three main features. **First**, instead of relying on experts to write prompts, we devise a method of generating prompts that is friendly to black-box LLMs. **Second**, due to the token limit of the LLM, we train a model to learn how to pair different inputs with appropriate demonstrations and prompts to alleviate the sensitivity problem of ICL. **Third**, during inference, we aim to maximize the LLM's ICL capability by exposing it to human labels via constructing an optimal subgraph of diverse demon-

strations/prompts. Based on the idea of ensemble learning, we request LLM multiple times with different inputs on the optimal subgraph and finally use the averaged score as the output of our dialogue evaluation system. We validate our method on five datasets and it exceeds all supervised and self-supervised learning baselines.

## 2 Related Work

**In-Context Learning** ICL is the process by which models learn from context examples during forward propagation, without parameter updates. ICL currently performs very well on traditional NLP tasks (Kim et al., 2022; Min et al., 2022), but ICL is still sensitive to how the prompts are written (Zhao et al., 2021), the selection and the order of demonstrations (Lu et al., 2022; Liu et al., 2022). For automatic labeling, ICL has shown good results and low cost on tasks such as word sense disambiguation, text summarization, and question generation (He et al., 2023; Wang et al., 2021), but using ICL for dialogue evaluation is rare.

**Auto Prompt Generation** One branch of prompt generation relies on paraphrasing of the seed prompt, including back-translation (Jiang et al., 2020), synonym substitution (Yuan et al., 2021), etc. Other methods use training data and labels to guide the prompt generation process, but they rely on the gradient of the model (Shin et al., 2020) or the confidence level of the output (Ding et al., 2022; Gao et al., 2021). Unlike the above approaches, we utilize the ability of LLM to learn from human demonstrations and induce appropriate prompts. Our approach is easy to use, intuitive, and well-adapted to black-box LLMs.

**Dialogue Evaluation** Most dialogue evaluation methods exploit the self-supervised learning capabilities of LLMs. A small amount of work design methods specifically around coherence (Huang et al., 2020; Ye et al., 2021), but most of the work centers around enabling a pre-trained LLM to perform multidimensional evaluation (Pang et al., 2020; Zhang et al., 2021b). Zhong et al. (2022) converts the fine-grained dimension evaluation into a unified Boolean QA problem, training T5 (Raffel et al., 2020) on four self-supervised tasks. Sinha et al. (2020) utilizes Noise Contrastive Estimation to train the model and construct a one-size-fits-all score. Whereas Mehri and Eskenazi (2020b) take the opposing view that the evaluation of dialogue is diverse and there is no one-size-fits-all metric,

thus proposing multiple interpretable sub-metrics. Zhang et al. (2022) propose a pairwise ranking loss to learn different sub-metrics, and integrate them by metric ensemble and multitask learning as the dialogue-level score. These works bring strong prior knowledge to the LLM via adjusting the training tasks and data, but none of them can be quickly adapted to new scoring criteria. Additionally, some works propose using the log probability of the LLM, such as BartScore (Yuan et al., 2021) and GPTscore (Fu et al., 2023), to evaluate the quality of dialogues. However, for certain evaluation dimensions, these methods rely on customizing specific task descriptions and prompt templates.

## 3 Problem Formulation

Our goal is to construct a dialogue evaluator that automates *human-like* scoring. We denote the input of the dialogue evaluator as $x$, which may contain three parts: the conversation context $c$ (usually contains several rounds of conversation), a response $r$ immediately following the context, and the conversation background $b$ (e.g., knowledge contained, persona information, etc.). Dialogue evaluation is categorized into dialogue-level evaluation, where the evaluator scores the overall context $c$ from different dimensions (e.g., topic depth, informative, etc.), and turn-level evaluation, where the evaluator scores the next round of response $r$ succeeding the context $c$ and the possible background $b$. Given an input $x$, the evaluator is expected to output a score $\hat{y}$ for a particular evaluation dimension. We adopt the reference-free setting for both turn-level and dialogue-level evaluation due to the one-to-many characteristic of reference-based methods (Zhao et al., 2017). Exhausting all potential responses for a reference is unfeasible.

We denote the labeled dataset as $\mathcal{D}_{train} = ((x_1, y_1), ..., (x_n, y_n))$, where $y$ is the average score of several human annotators. We define the set of inputs and labels for $\mathcal{D}_{train}$ as $\mathcal{D}_{train}^x$ and $\mathcal{D}_{train}^y$. Given $\mathcal{D}_{train}^x \sim \mathrm{P}_x$ and $\mathcal{D}_{train}^y$, we aim to automate the scoring process of another dataset $\mathcal{D}_{test}^x$ from the same distribution $\mathrm{P}_x$, ensuring as high consistency as possible between the model's scoring $\widehat{\mathcal{D}_{test}^y}$ and the humans' scoring $\mathcal{D}_{test}^y$.

## 4 Method

### 4.1 Dialogue Scoring via In-Context Learning

We employ a black-box LLM to score dialogues via in-context learning. We provide the LLM with

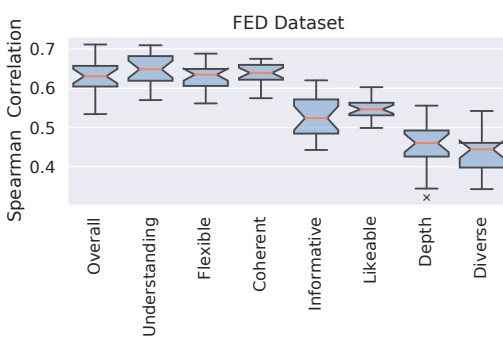

(a) The box plot of each evaluation dimension encompasses the outcomes derived from a set of 20 manually written prompts.

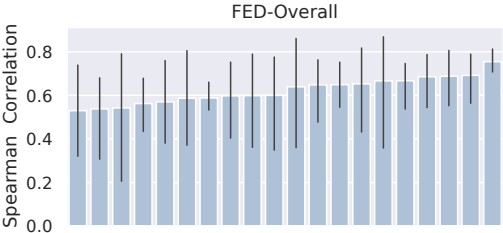

(b) Each column represents a different random seed for sampling the demonstrations. Error bars correspond to the results of five different test sets.

Figure 2: Results of requesting the LLM via ICL.

demonstrations $\mathbf{x}_{demo}$, the prompt $p$ describing the current task, and the input $x$. The demonstrations $\mathbf{x}_{demo}$ contains $k$ pairs of labeled samples $((x_{demo,1}, y_{demo,1})..., (x_{demo,k}, y_{demo,k}))$, and we denote $\mathbf{x}_{\tilde{demo}}$ as demonstrations without human scores. We have a template function $\tau$ to fill all inputs, where the filled template is denoted as $\tau(\mathbf{x}_{demo}, p, x)$. The exact format of template is as follows (see Table 9 for a complete filled template):

[$\mathbf{x}_{demo}$] [Task Prompt $p$] [$x$] Score:.

The template for demonstrations [$\mathbf{x}_{demo}$] is:

[$x_{demo,1}$] Score:[$y_{demo,1}$] ...

[$x_{demo,k}$] Score:[$y_{demo,k}$].

To obtain the dialogue evaluation score, we request the LLM with $\tau(\mathbf{x}_{demo}, p, x)$. We extract the first floating point number from the LLM's output as the predicted score $\hat{y}$. We define the LLM scoring function as $\mathcal{LLM} : \tau(\mathbf{x}_{demo}, p, x) \to \hat{y}$.

### 4.2 Inverting Prompts from Training Set

In practice, the variance of manually written prompts can be quite large (Figure 2(a)), and it may take us constant trial and error to manually find a suitable prompt. The whole process is inefficient and trivial. To reduce the manual workload

of composing prompts, we let the LLM invert suitable prompts by observing several pairs of labeled examples from $D_{train}$ at a time (limited by the context length of the LLM). The templates used for inversion are as follows:

[$\mathbf{x}_{demo}$] [Inversion Prompt ($p_{inver}$)]

Below we provide a filled template with a simple inversion prompt $p_{inver}$:

*[omitted few-shot demonstrations $\mathbf{x}_{demo}$]*

***Inversion Prompt:*** *Please design a prompt that will be used to request the language model to score the response. Examples of responses and scores are provided for reference. Prompt:*

For brevity, we omit the few shot demonstrations in the above example. For a complete filled template, please refer to Table 8. For more inversion prompts used in practice, please refer to Table 10.

Writing $p_{inver}$ still requires trial and error but instead of observing the entire training set, you simply verify whether the generated prompt is reasonable for the current evaluation dimension. (An inadequate inversion prompt may lead LLM to mimic the content in the demonstration, continue the inversion prompt, or produce content that is completely unrelated to the current evaluation dimension). In practice, we keep one optimal inversion prompt for each evaluation dimension.

In order to derive $\ell$ optimal prompts for the LLM to perform ICL for a dialogue scoring task, the initial step involves generating $m$ prompt candidates ($m$ is set to be 20 times $\ell$). To this end, we request the LLM with $m$ filled inversion templates (the inversion template function is denoted as $\tau_{inver}(\mathbf{x}_{demo}, p_{inver})$), each with a different set of few shot demonstrations $\mathbf{x}_{demo}$. The best $\ell$ prompts are selected according to the squared error $\mathcal{S}_e$ (lower is better) on the training set $\mathcal{D}_{train}$, which is defined as

$$\mathcal{S}_e = \sum_{i=1}^{|\mathcal{D}_{train}|} (\mathcal{LLM}(\tau(\mathbf{x}_{demo}, p, x_i)) - y_i)^2, \quad (1)$$

where the candidate prompt $p$ is the output of requesting the LLM with $\tau_{inver}(\mathbf{x}_{demo}, p_{inver})$ and few shot demonstrations $\mathbf{x}_{demo}$ are randomly selected for each $p$. We denote the set of selected prompts as $\mathcal{P}_{\mathcal{S}}$.

## 4.3 Search for the Best Combination of Demonstrations and Prompt

It is believed that ICL is sensitive to the prompt, the selection, and the order of in-context examples (Zhao et al., 2021) (proved by Figure 2(b) as well), it is not optimal to randomly select demonstrations for each $x$ and apply the same globally preferred prompt to all the samples, nor can we include the entire training set due to the token limit of the LLM. To resolve the uncertainty caused by random sampling, we opt to tailor the combination of demonstrations and the prompt for each input $x$. To this end, we think of mining the pairing knowledge directly from the training set $\mathcal{D}_{train}$.

### 4.3.1 Dataset Construction

We construct a training set $\mathcal{O}$ for training a selector $f_{selector}$ to choose the optimal prompt and few shot demonstrations for each input $x$. Specifically, we denote the training set as $\mathcal{O} = \{(x_1^o, y_1^o), ..., (x_i^o, y_i^o), ...(x_{n_o}^o, y_{n_o}^o)\}$, where $n_o$ is the size of the $\mathcal{O}$ and $x_i^o$ contains the input $x_i$ that needs to be scored, $k$-shot demonstrations $\mathbf{x}_{demo,i}$ and a task prompt $p_i$. The label $y_i^o$ represents the confidence level at which we should select $\mathbf{x}_{demo,i}$ and $p_i$ for the given $x_i$. When we feed the LLM with a higher-confidence combination (input as $\tau(\mathbf{x}_{demo,i}, p_i, x_i)$), the selector $f_{selector}$ sees that the LLM's scoring will be closer to the human scoring. For the exact calculation of $y_i^o$, please refer to Equation 6. For a detailed description of constructing $\mathcal{O}$, please refer to Algorithm 1.

---

**Algorithm 1:** Build the training set $\mathcal{O}$ for the demo and prompt selector $f_{selector}$.

---

**Input** : Training set $\mathcal{D}_{train}$; $\mathcal{LLM}$;
         Set of selected prompts $\mathcal{P}_{\mathcal{S}}$;
         Template function $\tau$;
         Number of repetitive sampling $\gamma$;
**Output** : Training Data $\mathcal{O}_x, \mathcal{O}_y$ for $f_{selector}$

$\mathcal{O}_x \leftarrow \{\}, \mathcal{O}_y \leftarrow \{\}$;
**for** $x_i, y_i \in \mathcal{D}_{train}$ **do**
    **for** $p_i \in \mathcal{P}_{\mathcal{S}}$ **do**
        **for** 1 **to** $\gamma$ **do**
            $\mathbf{x}_{demo} \leftarrow$ select $k$ random
            samples from $\mathcal{D}_{train} \setminus \{x_i, y_i\}$;
            $\mathcal{O}_x \leftarrow \mathcal{O}_x \cup \{(\mathbf{x}_{demo}, p_i, x_i)\}$;
            $\hat{y}_i \leftarrow \mathcal{LLM}(\tau(\mathbf{x}_{demo}, p_i, x_i))$;
            $\mathcal{O}_y \leftarrow \mathcal{O}_y \cup \{1/e^{|y_i-\hat{y}_i|}\}$;
        **end**
    **end**
**end**
**return** $\mathcal{O}_x, \mathcal{O}_y$

---

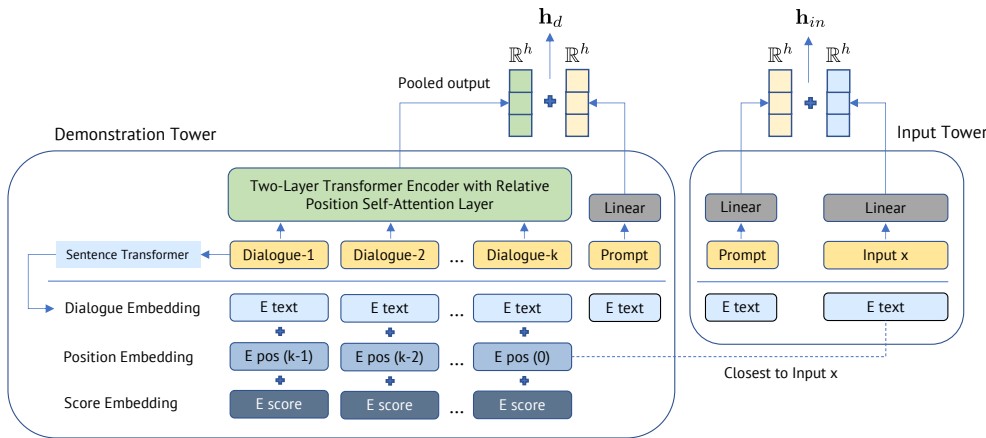

Figure 3: Main structure of $f_{selector}$, diagram of the input and output of the demonstration and input towers. The number in brackets after *E pos* is the input to the position embedding layer and represents the distance between the current example dialogue and the input $x$. The three linear layers are independent and do not share parameters.

### 4.3.2 Demonstration and Prompt Selector

We construct the dataset $\mathcal{O}$ to train a selector $f_{selector} : x^o \rightarrow y^o$. The model $f_{selector}$ is a two-tower model, where $\mathbf{x}_{demo}$ and $x$ are fed into two different towers, while the prompt $p$ is shared by both. Please refer to Figure 3 for an overview structure of $f_{selector}$.

**Demonstration Tower** The input of the demonstration tower $f_{demo}$ is $\mathbf{x}_{demo}$ and $p$. The $\mathbf{x}_{demo}$ consists of $k$ pairs of dialogues and scores, for which we do three levels of embedding extraction. Firstly, we use the Sentence Transformer [2] $f_{mpnet}$ to convert the score-free demonstration $\mathbf{x}_{\tilde{demo}}$ into an text embedding matrix: $\mathbf{E}_{text} = f_{mpnet}([x_{\tilde{demo},1}, ..., x_{\tilde{demo},k}]) \in \mathbb{R}^{k \times h}$. The exact form of input can be found in Table 7.

The second level joins the absolute position information (Devlin et al., 2019). We denote the absolute position embedding as $\mathbf{E}_{pos} = f_{pos}([pos_1, ..., pos_k]) \in \mathbb{R}^{k \times h}$, where $f_{pos}$ is the embedding layer. The number $pos_i$ refers to the distance between $x_{demo,i}$ and $x$ (for the closest position to $x$, the number $pos_i$ equals 0).

The third level is to add the scoring information. To obtain a legitimate score embedding, we bucket the whole scores $\mathcal{D}_{train}^y$ with Freedman Diaconis Estimator (Freedman and Diaconis, 1981). Each score $y$ is converted into an integer $\tilde{y}$. We calculate the score embedding as follows: $\mathbf{E}_{score} = f_{score}([y_{\tilde{demo},1}, ..., y_{\tilde{demo},k}]) \in \mathbb{R}^{k \times h}$, where $f_{score}$ is an embedding layer. Finally we obtain the input of the demo tower as $\mathbf{E}_{demo} = (\mathbf{E}_{text} + \mathbf{E}_{pos} + \mathbf{E}_{score}) \in \mathbb{R}^{k \times h}$.

Next we introduce the main function of the demonstration tower, which is denoted as $f_{demo} : (\mathbf{E}_{demo}, f_{mpnet}(p)) \rightarrow h_d \in \mathbb{R}^h$. It consists of an affine transformation $f_{l,1}$ for prompt embedding $f_{mpnet}(p)$ and a two-layer transformer encoder (Vaswani et al., 2017), with each transformer layer composed of a multi-head self-attention layer followed by a feed-forward network. It is worth noting that we replace the vanilla self-attention layer with a self-attention layer with relative position (Shaw et al., 2018) to capture the changes in $y^o$ caused by the different relative positions between the demonstrations. The output of the demonstration tower $f_{demo}$ is calculated as

$$\mathbf{h}_d = f_{pool}(f_{tran}(\mathbf{E}_{demo})) + f_{l,1}(f_{mpnet}(p)), \quad (2)$$

where $f_{tran}$ is the transformer encoder, $f_{pool}$ is the average pooling layer and $f_{l,1}$ is a linear layer.

We briefly discuss the reasons for introducing both the relative position self-attention layer and the absolute position encoding. Both of them can perceive the changes brought by different permutations of demonstrations, but the focus of the captured position information is different. We adopt the relative position layer to learn the effective order information among demonstrations, e.g. demonstration $x_{demo,a}$ needs to be before $x_{demo,b}$. But the two-tower architecture makes it impossible for the relative position layer to perceive the positional relationship between the demonstrations and the input $x_i$. Thus we adopt the absolute position encoding to allow the model to learn positional features when the distance relationship between demonstrations and the input $x_i$ matters. These two positional encodings are complementary.

---

[2]We use the all-mpnet-base-v2 model from www.sbert.net

**Input Tower** The structure of the input tower model $f_{input} : (f_{mpnet}(x), f_{mpnet}(p)) \rightarrow \mathbf{h}_{in} \in \mathbb{R}^h$ is simple, which consists of two linear layers $f_{l,2}$ and $f_{l,3}$, with the output being calculated as

$$\mathbf{h}_{in} = f_{l,2}(f_{mpnet}(x)) + f_{l,3}(f_{mpnet}(p)). \quad (3)$$

**Output and Loss Function** The output of $f_{selector}$ is calculated as

$$\hat{y}^o = \sigma(\mathbf{h}_d, \mathbf{h}_{in}) = \frac{1}{1 + \mathrm{e}^{-(\mathbf{h}_d \cdot \mathbf{h}_{in})}} \quad (4)$$

and the loss function is Mean Squared Error:

$$\mathrm{MSE} = \frac{1}{n}\sum_{i=1}^{n}(\hat{y}_i^o - y_i^o)^2, \quad (5)$$

where the ground truth label $y_i^o$ is calculated as:

$$y_i^o = \frac{1}{\mathrm{e}^{\left|y_i - \mathcal{LLM}(\tau(\mathbf{x}_{demo,i}, p_i, x_i))\right|}}. \quad (6)$$

We design our $f_{selector}$ as a two-tower structure for fast inference. The output of $f_{demo}$ depends only on the training set and can be calculated in advance so that the main inference process is the multiplication of two matrices. Even if the number of demonstration combinations reaches $100,000$ or more, the calculation can be done within a second.

### 4.4 Constructing the Optimal Subgraph of Demonstrations and Prompts

In practice, we find that for the same input $x$, multiple combinations of $\mathbf{x}_{demo}$ and $p$ have close high scores $\hat{y}^o$ at inference. Inspired by this phenomenon, we intend to request the LLM multiple times, each time with different examples from

---

**Algorithm 2:** Greedy Algorithm

**input** : Graph $\mathcal{G}$
$\quad\quad\quad$ Optimal subgraph $\mathcal{G}'^* = \{\}$, $n_{\mathcal{G}'}$
**output** : Optimal subgraph $\mathcal{G}'^*$

$v^* \leftarrow \arg\max_{v_i \in \mathcal{G}}(W_v(v_i));$
$\mathcal{G} \leftarrow \mathcal{G} \setminus \{v^*\};$
$\mathcal{G}' \leftarrow \mathcal{G}' \cup \{v^*\};$
**while** $|\mathcal{G}'^*| < n_{\mathcal{G}'}$ **and** $|\mathcal{G}| > 0$ **do**
$\quad$ $v^* \leftarrow \arg\max_{v_i \in \mathcal{G}} \mathcal{Q}(\mathcal{G}'^*, v_i);$
$\quad$ $\mathcal{G} \leftarrow \mathcal{G} \setminus \{v^*\};$
$\quad$ $\mathcal{G}'^* \leftarrow \mathcal{G}'^* \cup \{v^*\};$
**end**
**return** $\mathcal{G}'^*$

---

$\mathcal{D}_{train}$, and integrate multiple results at the end. At inference, given an input $x$, we randomly sample $\zeta$ different combinations of $\mathbf{x}_{demo}$ repeatedly for each prompt in $\mathcal{P}_{\mathcal{S}}$, forming a set of size $\zeta \times |\mathcal{P}_{\mathcal{S}}|$. We regard this set as an undirected fully connected graph $\mathcal{G}$. Specifically, we define $\mathcal{G} = (V, E, W_v, W_e)$, where each node $v \in V$ represents a pair of $\mathbf{x}_{demo}$ and $p$. The function $W_v : v \rightarrow y^o \in \mathbb{R}$ acquires the confidence value from $f_{selector}(\mathbf{x}_{demo}, p, x)$. We denote $W_e : e \rightarrow W_e(e) \in \mathbb{R}$ as an edge mapping function. The weighted edge $W_e(e_{ij})$ between nodes represents the discrepancy between node $v_i$ and node $v_j$, which is computed as follows,

$$W_e(e_{ij}) = 1 - \frac{|\mathbf{x}_{\widetilde{demo},i} \cap \mathbf{x}_{\widetilde{demo},j}|}{k}, \quad (7)$$

where $W_e(e_{ij})$ ranges between $[0-1]$, 1 means that $v_i$ and $v_j$ do not have any identical demonstrations, and 0 means that $v_i$ and $v_j$ share exactly the same demonstrations.

We aim to find an optimal subgraph $\mathcal{G}'^*$ such that the confidence scores ($y^o$) of the nodes in the optimal subgraph are high while satisfying that the discrepancy between nodes is as large as possible. We create a formula for scoring the subgraph $\mathcal{G}'$ when a new node $v_i$ is added:

$$\mathcal{Q}(\mathcal{G}', v_i) = \sum_{j=1}^{n_{\mathcal{G}'}}\left(\frac{2W_e(e_{ij}) \cdot W_v(v_i)}{W_e(e_{ij}) + W_v(v_i)}\right) \mid v_j \in \mathcal{G}'. \quad (8)$$

We denote $n_{\mathcal{G}'}$ as the size of subgraph $\mathcal{G}'$. Combining the above Equation 8 with the summation operation in the parentheses of Equation 9, we can use it to evaluate the overall quality of a subgraph. In the ideal situation, we can find the optimal subgraph based on the Equation 9, where $\Omega_{\mathcal{G}'}$ represents the sampling space of $\mathcal{G}'$ of size $n_{\mathcal{G}'}$.

$$\mathcal{G}'^* = \arg\max_{\mathcal{G}' \in \Omega_{\mathcal{G}'}}(\sum_{i=1}^{n_{\mathcal{G}'}} \mathcal{Q}(\mathcal{G}'_{\setminus v_i}, v_i), v_i \in \mathcal{G}') \quad (9)$$

Due to the NP-hardness of the problem[3], determining the globally optimal subgraph is not feasible within a finite time frame. Consequently, in practice, we resort to the Greedy Algorithm (See Algorithm 2). Once retrieved the optimal subgraph $\mathcal{G}'^*$, we iterate through all nodes of the optimal subgraph, obtain $n_{\mathcal{G}'}$ different sets of filled templates

---

[3]Our problem is a variant of the Maximum Weighted Clique problem (Östergård, 2001), which is NP-Hard.

$\tau(\mathbf{x}_{demo}, p, x)$, request the LLM, and obtain $n_{\mathcal{G}'}$ scores. We simply take the average score as the final score for a certain evaluation dimension.

## 5 Experiments

### 5.1 Datasets

We use five datasets for dialogue evaluation, including FED (Dialogue) (Mehri and Eskenazi, 2020a), Topical-USR, Persona-USR (Mehri and Eskenazi, 2020b), DailyDialog-Zhao and Persona-Zhao (Zhang et al., 2020) (See § A.1 for more details). The FED dataset is evaluated at the dialogue level while the other four are evaluated at the turn level. We choose specific evaluation dimensions for each dataset, as shown in Table 1 and Table 2. These evaluation dimensions are selected because they have high inter-annotator agreement (with an average Spearman's rank correlation coefficient of 0.79), which is a guarantee of label reliability and consistency.

### 5.2 Experimental Settings

Since all evaluation datasets contain very few samples ($< 1000$), we adopt the 5-fold cross-validation method, in each split, $\mathcal{D}$ is partitioned into a training set $\mathcal{D}_{train}$ and a test set $\mathcal{D}_{test}$. For each dimension, we report the mean value on the 5 test sets. As the LLM is extremely sensitive to the choice of demonstrations (Figure 2(b)), for those baselines that require random sampling of demonstrations, we repeat the experiments 4 more times (each time with a different random seed for sampling) and report the mean value of these experiments.

We set the number of demonstrations filling the inversion-template $\tau_{inver}$ to be 14. After inverting prompts from $\mathcal{D}_{train}$, we keep only four of the best prompts ($\ell$ is set to 4, thus $|\mathcal{P}_{\mathcal{S}}|$ is 4).

For training details, we set the learning rate, the batch size, and the epoch of $f_{selector}$ to $1e^{-4}$, 32, and 50 respectively. The hidden size $h$ for $f_{demo}$ and $f_{input}$ is set to 768. We chose Adam (Kingma and Ba, 2014) as our optimizer. In the process of constructing $\mathcal{O}$, the number of repetitive sampling $\gamma$ is set to 4; during inference, the number of repetitive sampling $\zeta$ for each selected prompt in $\mathcal{P}_{\mathcal{S}}$ is set to 512, rendering the size of $\mathcal{G}$ being $\zeta \times |\mathcal{P}_{\mathcal{S}}|$. We set the size of the optimal subgraph $n_{\mathcal{G}'}$ to 5.

The black-box LLM we request in our experiments is *gpt3.5-turbo-0301*[4], which is economical[5]

---

[4]https://platform.openai.com/docs/models/gpt-3-5
[5]Charge only 2\$ per $1,000,000$ token

yet powerful. We set the decoding temperature $t$ and the top-$p$ value (Holtzman et al., 2020) to 0.0 and 1.0 respectively.

#### 5.2.1 Baselines

- **D-score (Zhang et al., 2021a)** D-score is one of the best self-supervised turn-level evaluation frameworks. It devises a range of evaluation tasks and a multi-task learning framework. In order to maximize the performance of the D-score, we adopt the approach described in (Zhao et al., 2020a) and further fine-tune the D-score checkpoint on the target datasets.

- **FineD-Eval$_{mu}$ (Zhang et al., 2022)** We select FineD-Eval$_{mu}$ and it is the SOTA self-supervised method for dialogue-level evaluation. It tailors specific pre-training tasks to different evaluation dimensions, highlighting a metric ensemble method[6].

- **Ruber (Tao et al., 2018)** Ruber is one of the best supervised methods to perform dialogue evaluation, for fairness, we changed Ruber's backbone from RNN to Roberta Large (Liu et al., 2019). Ruber encodes context and responses separately, so we only experiment with Ruber on the turn-level evaluation datasets.

- **GTPScore (Fu et al., 2023)** GTPScore is a novel evaluation framework that leverages the emergent capabilities of pre-trained generative models to score generated texts in a training-free manner. For turn-level assessment, we use GTP-Score to evaluate the **overall** quality of the responses, while for dialogue-level **overall** assessment, we compute the GTPScore by averaging the scores obtained at the turn-level. We choose *text-davinci-003* from OpenAI for the experiment.

- **Fine-tuning of Pre-trained Language Models** Before the emergence of in-context learning capabilities for LLMs, most tasks in this area follow the paradigm of pre-training and fine-tuning. We select the Roberta-Large as the fine-tuning model, trained on the classification (based on the bucketed $\tilde{y}$) and regression tasks. The loss functions for classification and regression are Cross-Entropy and MSE. The input for supervised baselines is tokenized contexts and responses.

---

[6]Our results are different from those in the FineD-Eval paper due to the different experimental settings.

| | DDZ | PZ | UTC | | UPC | |
|---|---|---|---|---|---|---|
| | Appropriate. | Appropriate. | Ground. | Overall | Ground. | Overall |
| D-score | 0.523 | 0.555 | *0.327* | 0.478 | 0.389 | 0.501 |
| GPTScore (text-davinci-003) | - | - | - | 0.427 | - | 0.321 |
| Roberta-Large (cls) | 0.321 | 0.454 | 0.403 | *0.217* | 0.617 | *0.237* |
| Roberta-Large (reg) | 0.623 | 0.710 | 0.482 | 0.167 | 0.654 | 0.563 |
| Ruber-Roberta-Large | *0.202* | 0.580 | 0.352 | 0.497 | *0.174* | *0.211* |
| Human-P-Selected (z.s) | 0.662 | 0.678 | 0.507 | 0.474 | 0.579 | 0.459 |
| Human-P-Selected (5.s) | 0.684 | 0.679 | 0.482 | 0.527 | 0.690 | 0.494 |
| Auto-P-Random (5.s) | 0.585 | 0.589 | 0.460 | 0.525 | 0.645 | 0.571 |
| Auto-P-Selected (5.s) | 0.690 | 0.680 | 0.508 | 0.583 | 0.669 | 0.610 |
| Auto-P-Selected (5.s with BM25) | 0.576 | 0.643 | 0.523 | 0.616 | 0.658 | **0.653** |
| Auto-P-Selected-Ensemble (5.s) | 0.705 | 0.694 | 0.485 | 0.609 | 0.680 | 0.602 |
| ADOROR (ours, 5.s) | **0.733** | **0.711** | **0.529** | **0.623** | **0.703** | 0.637 |
| — w\o Optimal subgraph | 0.714 | 0.709 | 0.510 | 0.618 | 0.687 | 0.641 |

Table 1: Main results for turn-level evaluation. Statistically insignificant scores ($p > 0.05$) are italicized. The best results are in **bold** and the second best are underlined. 'P' stands for prompt. 'cls' and 'reg' stand for classification and regression. 'z.s' means zero-shot. '5.s' means five-shot. 'Appropriate.' and 'Ground.' are Appropriateness and Groundness (the degree to which the response uses knowledge from the background). 'DDZ', 'PZ', 'UTC', and 'UPC' are abbreviations for DailyDialogue-Zhao, Persona-Zhao, Topical-USR, and Persona-USR respectively. 'w\o Optimal subgraph' indicates that we choose the $\mathbf{x}_{demo}$ and $p$ with the highest $\hat{y_o}$ for inference.

| | Coh. | Div. | Dep. | Lik. | Und. | Fle. | Inf. | Overall |
|---|---|---|---|---|---|---|---|---|
| FineD-Eval$_{mu}$ | 0.604 | *0.357* | *0.363* | *0.374* | 0.389 | *0.382* | 0.442 | 0.392 |
| GPTScore (text-davinci-003) | - | - | - | - | - | - | - | *0.279* |
| Roberta-Large (cls) | *0.082* | *0.183* | 0.267 | *-0.032* | *0.082* | *0.185* | *0.159* | *0.047* |
| Roberta-Large (reg) | 0.299 | *0.125* | *0.280* | *0.165* | 0.261 | 0.326 | *0.285* | *0.258* |
| Human-P-Selected (z.s) | 0.622 | *0.260* | *0.235* | 0.545 | 0.439 | 0.529 | 0.409 | 0.590 |
| Human-P-Selected (5.s) | 0.632 | 0.432 | 0.451 | 0.534 | 0.635 | **0.617** | 0.516 | 0.616 |
| Auto-P-Random (5.s) | 0.612 | 0.366 | 0.451 | 0.482 | 0.590 | 0.514 | 0.510 | 0.613 |
| Auto-P-Selected (5.s) | 0.559 | 0.383 | 0.428 | 0.502 | 0.629 | 0.468 | 0.490 | 0.576 |
| Auto-P-Selected (5.s with BM25) | 0.599 | 0.352 | 0.430 | 0.475 | 0.533 | 0.451 | 0.513 | 0.591 |
| Auto-P-Selected-Ensemble (5.s) | 0.613 | 0.385 | 0.439 | 0.510 | 0.633 | 0.540 | 0.534 | 0.614 |
| ADOROR (5.s, ours) | **0.681** | 0.506 | **0.554** | **0.590** | **0.681** | 0.582 | **0.660** | **0.698** |
| — w\o Optimal Subgraph | 0.566 | **0.512** | 0.455 | 0.523 | 0.640 | 0.515 | 0.619 | 0.649 |

Table 2: Main results for the FED dataset. 'Coh.', 'Div.', 'Dep.', 'Lik.', 'Und.', 'Fle.', and 'Inf.' are abbreviations for Coherent, Diverse, Topic Depth, Likeable, Understanding, Flexible, Informative, and Overall, respectively.

- **Request LLM with Human Expert Prompts** We also test the results of requesting the LLM directly with an expert-written prompt (including the zero-shot and few-shot cases). For each evaluation dimension, we ask experts to write 20 prompts. we keep the best one for each dimension based on the prompt's performance on the training set (according to the Equation 1).

- **Request LLM with Inverse-Generated Prompts** This branch contains several variants: randomly sampling a generated prompt (Auto-P-Random, not evaluated on $\mathcal{D}_{train}$), choosing the prompt that gives the lowest square error on $\mathcal{D}_{train}$ (Auto-P-selected), using all prompts in $\mathcal{P}_{\mathcal{S}}$ (Auto-P-selected-Ensemble). For the ensemble variant, we request the LLM $\ell$ times for each sample and take the average score. Inspired by previous work (Rubin et al., 2022), we also test retrieving demonstrations with Okapi BM25 (Robertson et al., 2009).

If not specified, all baselines starting with Auto-P and Human-P, their $\mathbf{x}_{demo}$ is randomly sampled for each $x$ (same random seed for different methods).

### 5.3 Results and Analysis

We report the main results (Spearman's rank correlation coefficient with human labels) in Table 1 and Table 2, from which we can see that our ADOROR method is significantly better than all other baselines. It is also noticeable that ADOROR with-

|                                    | DDZ      | PZ       | UTC      | UPC      | FED      | Avg      |
| ---------------------------------- | -------- | -------- | -------- | -------- | -------- | -------- |
| w/o Relative Position Attention    | -2.046%  | 0.000%   | -0.562%  | -1.908%  | -8.164%  | -2.536%  |
| w/o Absolute Position Emebdding    | -1.637%  | -1.828%  | -3.387%  | 1.376%   | -6.171%  | -2.329%  |
| w/o Score Emebdding                | -2.093%  | -0.146%  | -1.850%  | -0.281%  | -7.238%  | -2.322%  |
| w/o Optimal Subgraph Construction  | -2.592%  | -0.281%  | -2.197%  | -0.824%  | -9.461%  | -3.071%  |

Table 3: The percentage of performance degradation without certain components for ADOROR.

|                    | Min     | Max     | Average |
| ------------------ | ------- | ------- | ------- |
| DDZ-Appropriate    | 1.57%   | 4.05%   | 2.89%   |
| PZ-Appropriate     | 1.40%   | 3.15%   | 2.24%   |
| UTC-Groundnes      | 0.68%   | 11.76%  | 5.62%   |
| UTC-Overall        | 0.53%   | 6.63%   | 3.25%   |
| UPC-Groundnes      | -0.94%  | 3.82%   | 1.56%   |
| UPC-Overall        | 2.77%   | 7.82%   | 5.62%   |
| FED-Coherent       | 7.24%   | 23.62%  | 14.04%  |
| FED-Diverse        | -3.61%  | 31.43%  | 14.10%  |
| FED-Depth          | -2.25%  | 26.23%  | 9.96%   |
| FED-Likeable       | 8.27%   | 24.97%  | 15.66%  |
| FED-Understanding  | 0.87%   | 14.73%  | 7.08%   |
| FED-Flexible       | -6.67%  | 22.43%  | 8.27%   |
| FED-Informative    | 4.75%   | 23.43%  | 17.25%  |
| FED-Overall        | 10.08%  | 14.12%  | 12.08%  |

Table 4: Sensitivity analysis for $\gamma$ and $\zeta$.

out constructing an optimal subgraph shows a decent improvement over Auto-P-selected and Auto-P-selected-Ensemble on almost all evaluation dimensions (except for Coherence, Flexible of the FED dataset), a result illustrates that finding the optimal prompt is not good enough, we need to more finely customize the demonstrations and the prompt for each input $x$, which emphasizes the need for training $f_{selector}$.

Comparing the expert hand-written prompts and the inverse-generated prompts, we find no significant differences, but the latter requires writing only one inversion prompt, which requires much less manual effort and relies less on expert experience.

For the traditional fine-tune paradigm, we see that Roberta-Large (reg) performs best, especially on turn-level datasets, but underperforms on dialogue datasets. In addition, MSE-based regression tasks are significantly better than cross-entropy-based classification tasks. For different sampling methods, we find that BM25-based retrieval does not have an advantage over random sampling.

### 5.4 Ablation Study

We report the results of the ablation study in Table 3, where we see that the construction of optimal subgraph at inference leads to the greatest improvement, and we believe that the fusion of the most suitable and diverse demonstrations/prompts makes greater use of the LLM's in-context learning capability. Other factors (including relative position attention layer, absolute position embedding, and score embedding) provide similar benefits. These results show it is beneficial to incorporate all these structures into ADOROR.

### 5.5 Sensitivity Analysis for Hyperparameters

In Table 4 we analyze the sensitivity of the hyperparameters $\gamma$ and $\zeta$, for which we perform a grid search, with $\gamma$ and $\zeta$ searches in the ranges (2,4,8) and (256,512,1024) respectively. We report the improvements of ADOROR relative to Auto-P-Selected-Ensemble. Min, Max, and Average correspond to the minimum, maximum, and mean values of the percentage of exceeding Auto-P-Selected-Ensemble under all combinations of $\gamma$ and $\zeta$. For a more detailed comparison, please refer to Figure 7, Figure 8 and Figure 9. This result proves that training $f_{selector}$ and building the optimal subgraph at inference are relatively robust.

We also test the sensitivity of ADOROR to different few-shot $k$ and conclude that ADOROR consistently outperforms the Auto-P-Selected method regardless of the few-shot variation, see Figure 10 for detailed experimental results.

### 6 Conclusion

Our automated dialogue scoring method allows the LLM to learn from *human labels* using in-context learning (ICL) in three stages. In the first stage, we let the LLM observe human scoring examples, from which the LLM induces suitable prompts for dialogue scoring. Additionally, the order and selection of demonstrations are crucial for ICL, and therefore, we train a demonstration/prompt selector using a new training set created by comparing the LLM and the human scoring. During inference, we provide the LLM with as diverse and suitable demonstrations and prompts as possible. Our ADOROR approach is flexible and adaptable to new evaluation dimensions or criteria, requiring only a small amount of human-labeled data.

## Limitations

- **Experimental Dataset** Regarding the datasets, the dialogues used for evaluation mostly originated from models such as Transformer (Vaswani et al., 2017), LSTM (Graves and Graves, 2012), GPT-2 (Radford et al., 2019), and others, which are much weaker than the current state-of-the-art LLMs. If our method can precisely imitate human scoring when annotating the dialogue history of more powerful chatbots, it would further validate the effectiveness of our approach. To accomplish this, we are collecting annotations from a fresh set of dialogue data generated by the strongest language models via crowdsourcing. We intend to incorporate additional annotation dimensions, such as persona consistency, verbal authenticity, etc. in the near future, as well as expand our experiments.

- **Choice of Large Language Model** We have solely experimented with gpt-3.5-turbo for automatic dialogue evaluation in great detail. For other LLMs, we also test how GPT-4 (OpenAI, 2023) performs on the PersonaZhao dataset. ADOROR achieves a Spearman rank's correlation coefficient of $0.817$, which is $4.2\%$ higher than the Auto-P-Selected baseline. This experimental result preliminarily demonstrates that our method has good LLM adaptability. Due to high costs, the non-availability of open APIs, and the limited capabilities of the language model itself, we have not tested more LLMs with comprehensive experiments, such as llama (Touvron et al., 2023), bard (Rahaman et al., 2023), and so on. As our approach is LLM-agnostic, we anticipate that future researchers can explore more LLMs.

- **Method** We did not conduct additional experiments to confirm the efficacy of our approach on white-box LLMs. Our method is tailored for black-box LLMs, but we anticipate that training ADOROR's demo selector and ensemble on an optimal subgraph could also enhance white-box LLMs in dialogue scoring (White-box denotes the model that allows access to gradients and confidence score distribution).

In addition, to make our approach lighter, we chose to extract an off-the-shelf, fixed-dimension embedding of the dialogue using the Sentence Transformer. This allows us to compress an arbitrary dialogue of fewer than 512 tokens into a vector of dimension 768, facilitating the expansion of the number of demonstrations. However, this approach reduces the representational power of the model. We may need to fine-tune the structure that encodes each dialogue to learn how to select the most appropriate demonstrations for the LLM. One way is to view $k$ demonstrations as a whole for tokenization and then feed $k$ tokenized $\mathbf{x}_{demo}$ into a LLM that supports modeling long sequences.

Regarding the way in which the LLM is requested, we currently require the LLM to output a score directly. However, on other tasks, a number of works have proposed adding a Chain-Of-Thought (COT) process to the LLM to allow the model to have an explicit thought process before output (Wei et al., 2022). We do not introduce COT into our experiments for two reasons: firstly, the evaluation datasets do not contain human reasons for scoring. Second, we did some preliminary experiments and found that using COT instead reduced the correlation between model scoring and human scoring. Why COT is not effective for the task of dialogue evaluation remains to be investigated.

## Ethics Statement

Our method pertains to ethics in two ways. Firstly, concerning data confidentiality, we adopt a powerful third-party LLM to analyze the dialogue data. It may be necessary to de-identify it to avoid privacy violations. The second aspect concerns the LLM's inherent bias. The LLM may exhibit biases towards gender, ethnicity, religion, or specific ideologies (Ray, 2023). The LLM's bias mainly stems from its training data, which is challenging for us as users to eliminate. Even after utilizing our automated scoring method, it is necessary to manually inspect the model's output to ensure that it is unbiased in particularly sensitive situations.

## Acknowledgement

Our work is supported by the Key Research and Development Program of Zhejiang Province (No. 2022C01011). We would like to express our special thanks to the four reviewers on OpenReview for their valuable comments. Currently, we are applying our ADOROR method in practice for evaluating Chinese dialogues, and we would like to extend our special thanks to **the NetEase crowd-sourcing platform**[7] for their strong support.

---

[7]https://fuxi.163.com/productDetail/17

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

# A Appendix

In the appendix, we provide more details of the dataset in §A.1. In addition, we provide a specific example of how to request the LLM to output an evaluation score in Table 9. We also provide the inversion prompts we use in our experiments in Table 10 and some examples of automatically generated prompts in Table 11.

Regarding the experimental results, we provide the correlation scores of different prompts on the training and test sets in Figure 4, Figure 5 and Figure 6, exploring whether prompts with better correlation scores on the training set can generalize better on the test set. In Figure 7, Figure 8, and Figure 9, we provide more detailed data for the sensitivity analysis of hyperparameters. In Figure 10, we show that our ADOROR method is consistently better than the Auto-P-Selected method for all few shot $k$ setting. For other baseline training details, please refer to §A.2. In addition, we provide hypothesis testing for our method in §A.3, which statistically demonstrate the effectiveness of our core innovation.

## A.1 Details of the evaluation datasets

For more detailed statistics on the datasets, please refer to Table 5.

- **DailyDialog-Zhao** Zhao et al. (2020b) created an evaluation dataset called DailyDialog-Zhao, which was based on 100 dialogues randomly selected from the test set of the DailyDialog corpus (Li et al., 2017). DailyDialog-Zhao assesses four criteria: appropriateness, language usage, relevance, and content. Each context-response pair is evaluated by four annotators using a 5-point Likert scale. After removing outliers, the inter-annotator agreement for appropriateness was measured using Krippendorff's alpha and found to be above 0.8.

- **Persona-Zhao** Gupta et al. (2019) created an evaluation dataset called Persona-Zhao, which was constructed in a similar way to the DailyDialog-Zhao dataset. The context-response pairs used in Persona-Zhao were obtained from dialogues in the test set of the PersonaChat corpus (Zhang et al., 2018). In Persona-Zhao, only the appropriateness quality was annotated, and the inter-annotator agreement for this category is above 0.8, as measured by Krippendorff's alpha.

- **Topical-USR and Persona-USR** Mehri and Eskenazi (2020b) created two human evaluation datasets called Topical-USR and Persona-USR. Both datasets use the same annotation scheme, where three dialogue researchers evaluate each context-response pair using six different categories to assess dialogue quality: Understandability (scored 0-1), Naturalness (scored 1-3), Context Maintenance (scored 1-3), Interest (scored 1-3), Knowledge Usage (scored 0-1), and Overall Quality (scored 1-5). The inter-annotator agreements for each category were measured using Spearman's rank correlation coefficient. For Topical-USR, the scores are 0.5102, 0.4871, 0.5599, 0.5811, 0.7090, and 0.7183, while for Persona-USR, the scores are 0.2984, 0.4842, 0.6125, 0.4318, 0.8115, and 0.6577.

- **FED (Dialogue)** FED dataset contains 124 conversations, with 40 conversations being generated by Meena, 44 by Mitsuku, and another 40 being human-human conversations (Mehri and Eskenazi, 2020a). These conversations were evaluated using 11 different criteria, with a high level of agreement among the evaluators (Spearman correlations ranging from 0.5 to 0.84).

## A.2 Details regarding other baselines

D-score has three models trained on different datasets: DSTC6, DSTC7, and PERSONA-CHAT. We pick the checkpoint of DSTC7 for further fine-tuning. For FineD-Eval, we pick the model trained on convai2 for evaluation. When evaluating the **Overall** dimension using GPTScore, the task description and the aspect definition are omitted. For supervised baselines using Roberta-Large, the labels for the classification task are obtained by bucketing the scores with the Freedman Diaconis Estimator (Freedman and Diaconis, 1981). For training of these supervised baselines, we split 10% of $\mathcal{D}_{train}$ into the validation set $\mathcal{D}_{val}$. The model with the lowest validation set is evaluated on the test set $\mathcal{D}_{test}$. We set epoch, batch size, and the gradient clipping value to 20, 16, and 1.0 respectively. We adjust the learning rate for each model to ensure that the model's loss on the training set converges consistently to below 0.5. We set a learning rate of $5e^{-6}$ for RUBER-base models and $1e^{-5}$ for the other models. The input of supervised baselines is tokenized contexts and responses.

| DailyDialog-Zhao | |
| --- | --- |
| #Instances | 900 |
| Avg.#Utts. | 4.7 |
| Avg.#Ctx/Hyp Words | 47.5 / 11.0 |
| Type | Turn |
| #Annotations | 14,400 |
| Models | LSTM Seq2Seq, Random and GPT-2 |
| Persona-Zhao | |
| #Instances | 900 |
| Avg.#Utts. | 5.1 |
| Avg.#Ctx/Hyp Words | 48.8 / 11.5 |
| Type | Turn |
| #Annotations | 3,600 |
| Models | LSTM Seq2Seq and GPT-2 |
| Topical-USR | |
| #Instances | 360 |
| Avg.#Utts. | 11.2 |
| Avg.#Ctx/Hyp Words | 236.3 / 22.4 |
| Type | Turn |
| #Annotations | 6,480 |
| Models | Transformers |
| Persona-USR | |
| #Instances | 300 |
| Avg.#Utts. | 9.3 |
| Avg.#Ctx/Hyp Words | 98.4 / 12.0 |
| Type | Turn |
| #Annotations | 5,400 |
| Models | Transformer Seq2Seq, LSTM Memory Network |
| FED (Dialogue) | |
| #Instances | 125 |
| Avg.#Utts. | 12.7 |
| Avg.#Ctx/Hyp Words | 113.8 / - |
| Type | Dialogue |
| #Annotations | 6,720 |
| Models | Meena, Mitsuku |

Table 5: Details of five evaluation datasets. 'Ctx' and 'Hyp' indicate dialogue context and model hypothesis. 'Utts' indicates utterance.

## A.3 Significance tests of ADOROR

Our core contribution is the proposal of two modules—training demonstration/prompt selector and constructing the optimal subgraph. Here we provide additional results of significance tests to demonstrate the effectiveness of these two modules. We use the Wilcoxon Sign Rank test (Woolson, 2005) to determine if there are significant differences between any two methods, with the calculation function denoted as $f_w(\cdot, \cdot)$. We represent the results of any method $m$ as $R_m = (r_1, \ldots, r_{|R|})$, where $r_i$ represents the the value of Spearman's correlation coefficient between a method's ratings of and human ratings on a test set, and $|R|$ equals

5 times the number of evaluation dimensions of all concerned datasets (multiply by 5 because we perform 5-fold cross-validation). Given two methods, A and B, their Wilcoxon Sign Rank test result ($p$-value) can be represented as $f_w(R_A, R_B)$. We present the Significance test results in Table 6. From this table, we can see that constructing the optimal subgraph steadily improve the performance and the improvement becomes more pronounced when we integrate both modules.

| | FED | UTC | UPC | Average |
|---|---|---|---|---|
| ADOROR w\o Optimal subgraph | 10.56% (0.009) | 2.13% (0.074) | 1.00% (0.139) | 4.56% (0.0002) |
| Auto-P-Selected (5.s) | 22.72% (0.0001) | 5.6% (0.036) | 4.7% (0.028) | 11.01% (5.542e-07) |

Table 6: Average improvement of **ADOROR** over **ADOROR w\o Optimal subgraph** and **Auto-P-Selected (5.s)** with few shot $k$=5. In parentheses are the $p$-values of Wilcoxon Sign Rank test.

| With Fact | Context:{context}\nFact:{Fact}\nResponse:{response} |
|---|---|
| With Persona | Context:{context}\nPersona:{Persona}\nResponse:{response} |
| without Fact or Persona | Context:{context}\nResponse:{response} |
| {context} example | User1: hi do you watch espn ?
User2: hi . yes i do sometimes . they cover some nba games .
User1: yes , i am a huge bucks fan . did you see ...
User2: no i did n't . what did he say ?
User1: that they should n't consider signing him . i agree .
User2: what made him such a bad player last year ?
User1: he did n't do much for the team and was overpaid . |
| {Fact} example | reggie miller , nba hall of famer ,
could n't escape his older sister cheryl 's shadow while in high school .
on january 26 , 1982 , he scored a career high 40 points .
he tried bragging about it on the car ride home ,
only to find out cheryl scored 105 points and broken 8 national records ! |
| {Persona} example | your persona:
i have a children and a dogs.
i own a house in florida.
i enjoy american sports.
i am a male. |
| {response} example | User2: i 'm not sure . i 'm sure there are some violin out there . |

Table 7: In the top column, we present the input form of Sentence Transformer, where we feed the entire multi-round conversation (context) along with the response into Sentence Transformer, outputs a vector of fixed length (768), as the dialogue embedding. For the USR dataset, the input context contains extra Fact or Persona.

```
**************************************************************************
Context:
User1: congratulations ! are you excited or nervous ?
User2: i'm excited as my cousin is getting married . do you study or work ?
User1: i do both . not tonight though .
User2: wow so good for you . yes is weekend enjoy it .
User1: i work in the morning , as a tour guide .
Response:
User2: yea i rock skinny jeans and leggings you
appropriateness Score: 1.5 [END]
**************************************************************************
Context:
User1: awesome ! i am going to college to become a physical therapist .
User2: same when i can get my parents of my back
User1: we should meet sometime ! after college !
User2: yes sounds like a great plan
User1: are you still in high school ?
Response:
User2: i am good , that is pretty cool . where do you work ?
appropriateness Score: 2.5 [END]
___________________________________________________
```

Your objective is to construct a task prompt that can generate an appropriateness score based on the given context and response. The score will determine the quality of the response in terms of appropriateness, and will fall within the range of (1.0, 5.0). The prompt should include the score range. In order to accomplish this, please carefully review the provided few shot demonstrations, note in particular how the above example is scored based on context and response. Please include as much detail as possible in the prompt. Please DO NOT include Context and Response in the task prompt.

The created task prompt will be utilized in the following manner during inference:

(few shot demonstrations)

(prompt)

task prompt:

Table 8: A concrete example of a filled inversion template $\tau_{inver}$ requesting the LLM. Here the number of demonstrations is set to 2. The evaluation dimension is 'appropriateness' for the DailyDialogue-Zhao dataset.

```
********************************************************************
Context:
User1: what kind of job do you have?
User2: i walk dogs to pay for college and food
User1: that is not such a bad job. it sounds fun.
User2: it is not stable enough and it does not pay enough
User1: but at least you get lots of fresh ai.
Response: User2: yes that's true and i love fresh air

appropriateness Score: 4.25 [END]
********************************************************************
Context:
User1: i'm a twitch streamer and famous at it
User2: that sounds like a the easiest money every
User1: very easy but i am allergic to water .
User2: that must be the worst User1: i'm a recluse . what about you ?
Response:
User2: i'm a chef . i love to cook .

appropriateness Score: 3.25 [END]
───────────────────────────────────────────────────
```
Please rate the appropriateness of the response on a scale of 1.0 to 5.0 based on the given context. The appropriateness score should reflect how well the response aligns with the context and the conversation as a whole. Please consider factors such as relevance, coherence, and tone when assigning the score.
```
********************************************************************
Context:
User1: yes , our shop sells watches . i help with repairs .
User2: very nice . maybe i'll stop in sometime . User1: i would like to meet you !
do you enjoy shopping ?
User2: i do enjoy shopping , as i think many women do . i must get going now .
User1: it was nice talking to you !
Response: User2: same to you . have anyone gone with your friend outside ?

appropriateness Score:
```

Table 9: A concrete example of a prompt requesting the LLM, which returns the appropriateness score. Here the few-shot $k$ is 2 and the dataset is Persona-Zhao. [END] is the end symbol for LLM. We can see that the demonstration is separated by a separator (*) and the prompt is also separated by a separator (-). In practice, we find that adding separators improves the performance of the LLM on in-context learning.

**Topical-USR and Persona-USR**:
Create a prompt that can generate a score between {Score Range} based on the {Evaluation Dimension} of a given Response to a {Background Type} and Context. This prompt is solely for evaluating an existing dialogue system during inference, and not for chatbot development or conversation generation. Review the few shot demonstrations provided and note how the example is scored based on the {Background Type}, Context, and Response. Do not include the {Background Type}, Context, or Response in the prompt. Provide as much detail as possible.
prompt:

**DailyDialogue-Zhao and Persona-Zhao**:
Your objective is to construct a task prompt that can generate an {Evaluation Dimension} score based on the given context and response. The score will determine the quality of the response in terms of {Evaluation Dimension}, and will fall within the range of {Score Range}. The task prompt should include the score range. In order to accomplish this, please carefully review the provided few shot demonstrations, note in particular how the above example is scored based on context and response. Please include as much detail as possible in the task prompt. Please DO NOT include Context and Response in the task prompt. The created task prompt will be utilized in the following manner during inference:
(few shot demonstrations)
(task prompt)
task prompt:

**FED**:
The dataset to be evaluated includes sets of conversations between a human and two open-domain dialog systems, Meena and Mitsuku2. In addition, the dataset also includes human-human dialogues collected in the same environment where one of the humans was selected to play the role of the system. You task now is to create a prompt that evaluates the {Evaluation Dimension} of a dialogue system's responses to a user's inputs, with a score within {Score Range}. Use the provided few-shot examples to guide the prompt creation in a way that generalizes beyond them. The prompt is solely for evaluating an existing dialogue system during inference, NOT for chatbot development or conversation generation. Avoid any user-system conversations in the prompt. DO NOT include system and user in the prompt! The created task prompt will be utilized in the following manner during inference:
(few shot demonstrations)
(task prompt)
prompt:

Table 10: Inversion prompts $p_{inver}$ used in our experiments. {Score Range}, {Evaluation Dimension}, {Background Type} are placeholders for the score range of the current evaluation dimension, the name of the evaluation dimension (e.g. Coherent), and the type of background knowledge, respectively. There are two types of background knowledge in the USR dataset, Fact and Persona.

**DailyDialogue-Zhao Appropriateness**:
Please rate the appropriateness of the response on a scale of 1.0 to 5.0 based on the given context. The appropriateness score should reflect how well the response fits the context and the conversation flow. A score of 1.0 indicates a highly inappropriate response, while a score of 5.0 indicates a highly appropriate response. Please consider the relevance, coherence, and tone of the response when assigning a score.

**Persona-Zhao Appropriateness**:
Given a context and response, rate the appropriateness of the response on a scale of 1.0 to 5.0, with 1.0 being completely inappropriate and 5.0 being highly appropriate. Consider the relevance of the response to the context, the coherence of the response, and the overall tone and language used in the response.

**Topical-USR Groundness**:
Based on the given Persona and Context, how well does the Response demonstrate an understanding of the Persona's interests, background, or experiences? Rate the Response on a scale of 0.0 to 1.0, where 0.0 indicates no understanding and 1.0 indicates a strong understanding. Consider the Persona's stated hobbies, family background, job, and other relevant details.

**Persona-USR Overall**:
Please rate the quality of the response on a scale of 1.0 to 5.0, with 1.0 being poor and 5.0 being excellent. Consider the coherence, relevance, and naturalness of the response in relation to the given context and persona. Use your best judgement to evaluate the quality of the response.

**FED Coherent**:
Evaluate the coherence of the dialogue system's responses to a user's input, with a score within (0.0, 2.0). A coherent response should be relevant, informative, and logically connected to the previous input. A score of 2.0 indicates a highly coherent response that effectively addresses the user's input, while a score of 0.0 indicates a completely incoherent response that is irrelevant or nonsensical. Use the provided few-shot examples to guide your evaluation.

**FED Topic Depth**:
Evaluate the depth of the dialogue system's responses to the user's input, with a score within (0.0, 2.0). The depth score measures the level of engagement, coherence, and relevance of the system's responses to the user's input. A score of 0.0 indicates a lack of engagement, coherence, and relevance, while a score of 2.0 indicates a high level of engagement, coherence, and relevance. The system's responses should demonstrate an understanding of the user's input, provide relevant information, and maintain a coherent and engaging conversation. Use the provided few-shot examples as a guide for evaluation.

Table 11: We present some top-performing prompts generated by requesting the LLM with the inversion prompt. These prompts are filtered based on proximity to human labels on $D_{train}$.

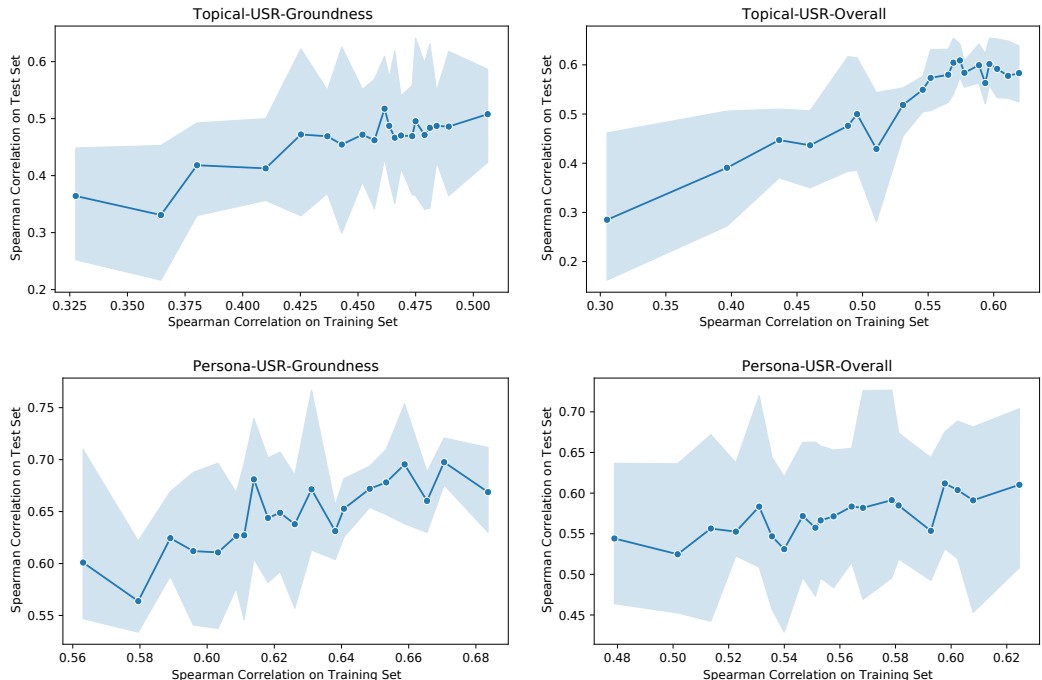

Figure 4: Each point corresponds to a prompt in $P_S$. With each prompt, we traverse the complete training and test sets, each time requesting LLM with the demonstration, prompt, and input that has been filled in the template. Based on the requested set of evaluation scores, we calculate the corresponding Spearman's rank correlation coefficient. The X-axis represents the correlation on the training set and the Y-axis represents the correlation on the test set. We find a clear pattern in the USR data set, where prompts with better correlation in the training set also correlates better with Human in the test set.

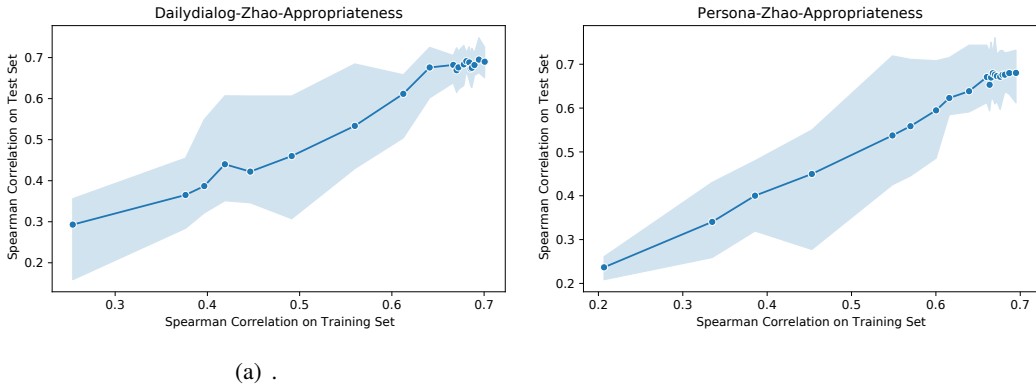

(a) .

Figure 5: The Dailydialog-Zhao and Persona-Zhao datasets: association of Spearman's rank correlation coefficient on the training and test sets. We can see a linear relationship between the Spearman's rank correlation coefficient of the test and training sets, with a slowing upward trend in the tail of the x-axis, but a very stable overall trend.

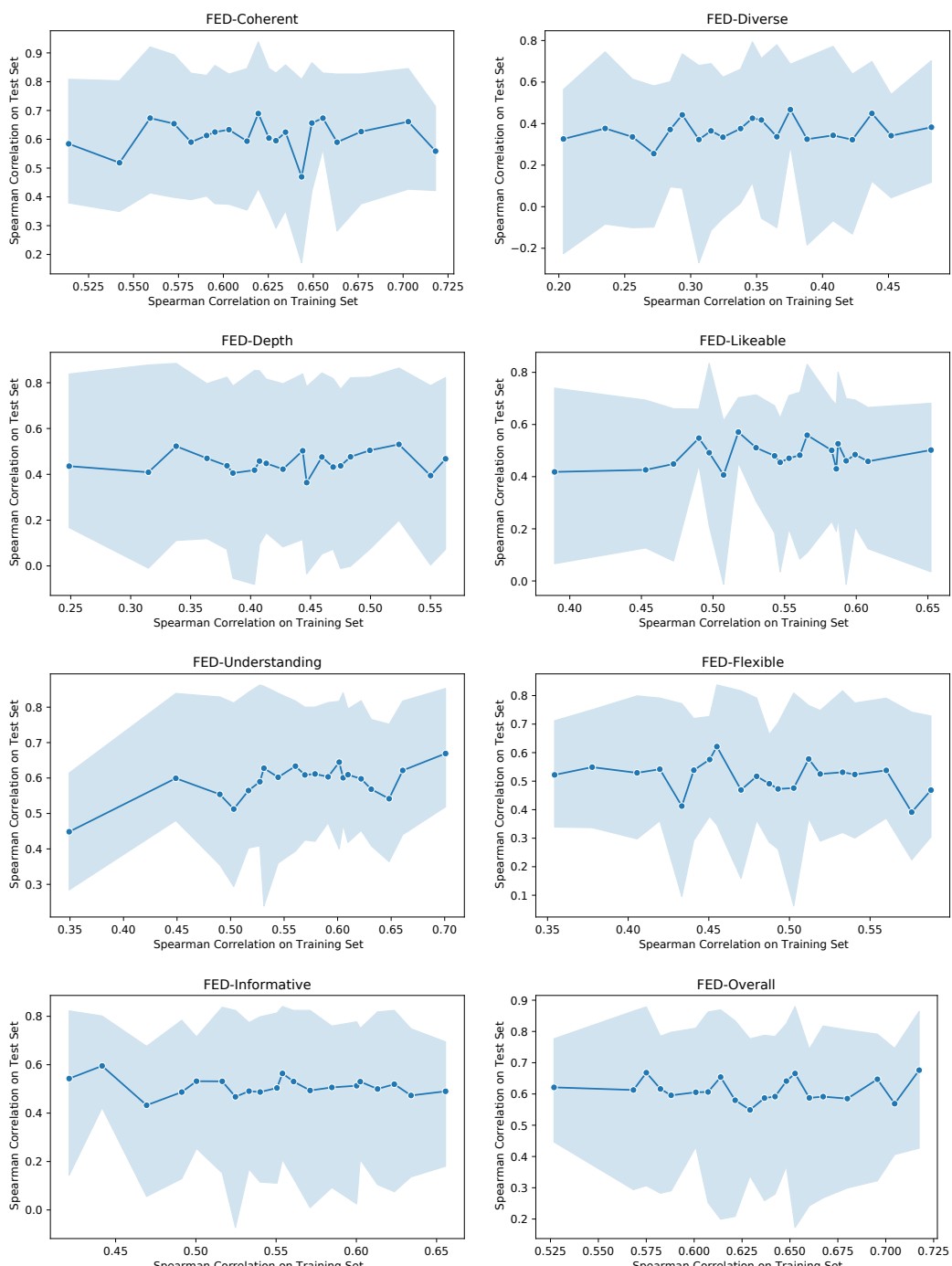

Figure 6: The FED dataset: association of Spearman Correlation on the training and test sets. We can see that except for the evaluation dimension of Understanding, the trend of the other evaluation dimensions is not obvious, i.e. filtering based on the performance of the prompt on the training set alone does not guarantee that we obtain the optimal prompt.

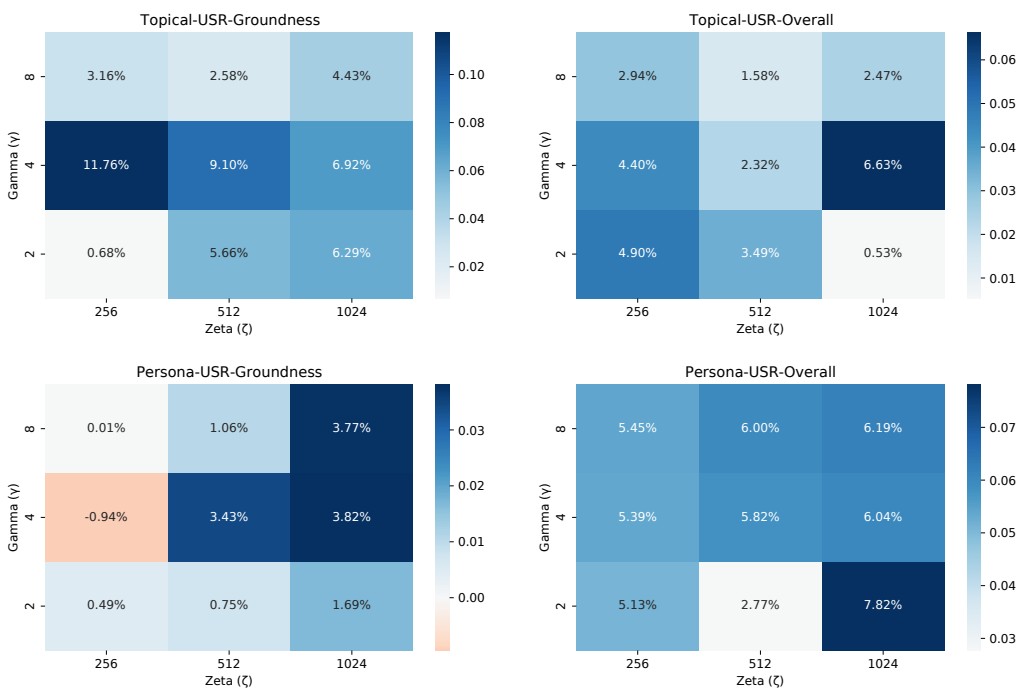

Figure 7: Sensitivity analysis of $\gamma$ and $\zeta$ on the USR dataset.

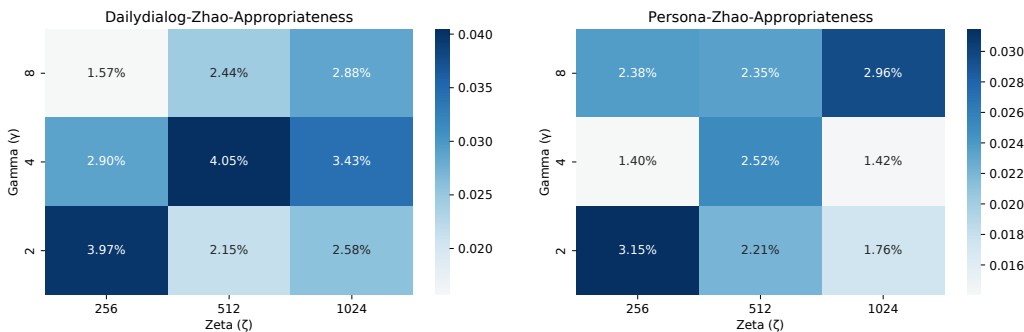

Figure 8: Sensitivity analysis for $\gamma$ and $\zeta$ on the Dailydialog-Zhao and Persona-Zhao datasets.

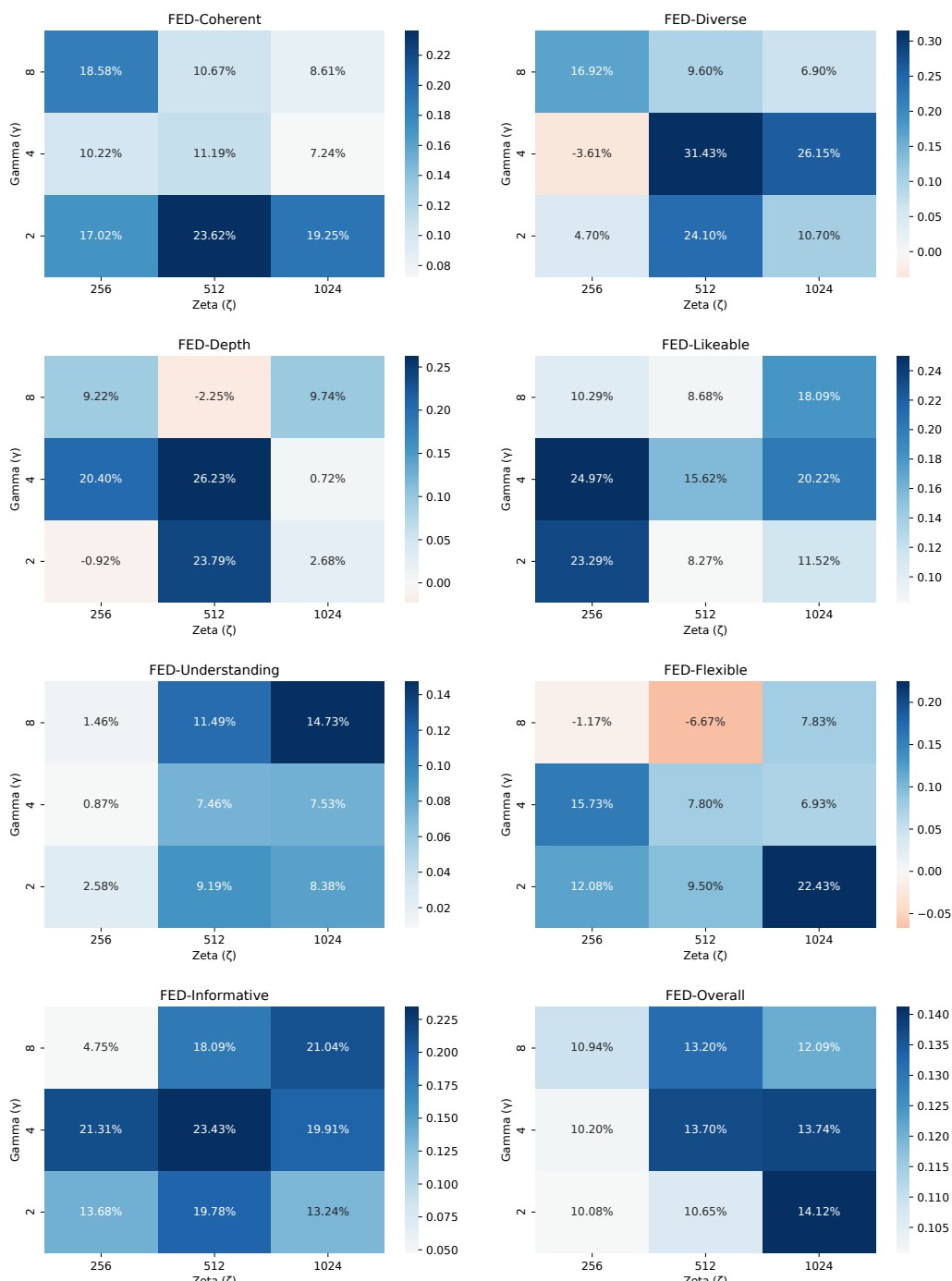

Figure 9: Sensitivity analysis for $\gamma$ and $\zeta$ on the FED dataset.

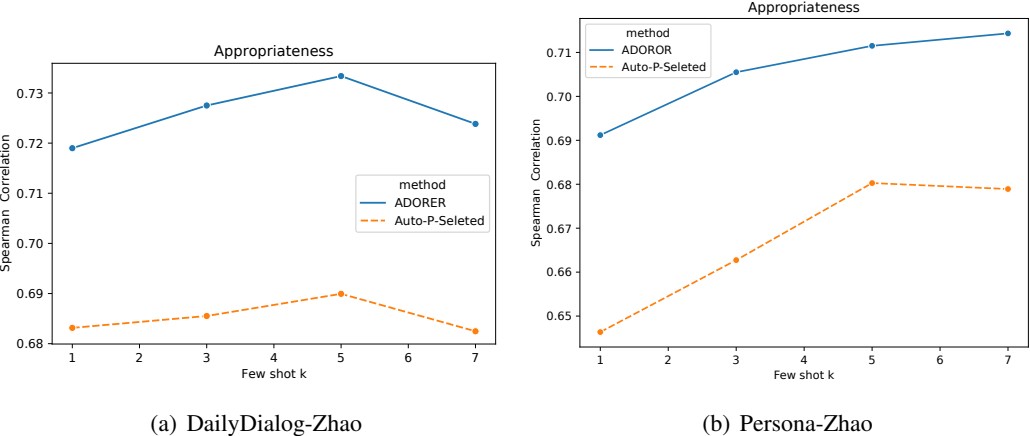

(a) DailyDialog-Zhao             (b) Persona-Zhao

Figure 10: Sensitivity analysis for few-shot $k$ on the DailyDialog-Zhao and Persona-Zhao dataset. We conclude that the ADOROR method is consistently better than the Auto-P-selected baseline, regardless of the number of demonstrations.