# OpenReview forum: "Just Adjust One Prompt: Enhancing In-Context Dialogue Scoring via Constructing the Optimal Subgraph of Demonstrations and Prompts"
_EMNLP/2023/Conference — EMNLP 2023 Main_

### Official Review · Reviewer_yAzT · 2023-08-05

**Soundness:** 4

**Excitement:**

4: Strong: This paper deepens the understanding of some phenomenon or lowers the barriers to an existing research direction.

**Paper Topic And Main Contributions:**

This paper proposes a systematic method that leverages in-context learning of LLM for dialogue evaluation in both dialogue-level and turn-level on multiple dimensions. The method includes inverting prompts from the training set, training a prompt selector, and finding an optimal subgraph during the inference stage. The experiments are done on five datasets and compared with multiple baselines.

**Questions For The Authors:**

* L186-188: I haven’t got the idea of “aim to automate the scoring process of another dataset $D^x_test$ from the same distribution $P_x$”. Could you explain more about why the goal is to test a dataset from the same distribution? I guess what you mean for the distribution $P_x$ is just a concept, an underlying distribution of real data, but not the distribution of $D^x_train$, right? That is, you split a collected dataset into the training and testing sets and say that they are from the same distribution, right? By the way, should the notation $D^test_y$ be changed to $D^y_test$ to be consistent with other notations?
* Experimental Question: Section 5.2 you mention “We set the decoding temperature and the top-p value to 0.0 and 1.0 respectively.” Just to confirm, does it mean you are using Argmax (since temperature = 0) and also not top-p sampling (since top-p value = 1)? Otherwise, I guess you want to mean you were using temperature=1.0 with just softmax sampling?


**Reasons To Accept:**

* The paper is full of information and well constructed.
* The in-context learning of LLM for dialogue evaluation is a timely method to explore.
* The method development and experiments have been carefully designed and are sound.


**Reasons To Reject:**

Minor:
* The method still looks time-consuming. Could you provide a detailed inference time comparison with other methods? It can be interesting to see the tradeoff between score correlation improvement and the extra consumed resources.
* The experiments can have a general NLG evaluation metric for reference for the overall score, such as BARTscore, SEScore, GPTscore etc.


**Reproducibility:**

3: Could reproduce the results with some difficulty. The settings of parameters are underspecified or subjectively determined; the training/evaluation data are not widely available.

**Reviewer Confidence:**

3: Pretty sure, but there's a chance I missed something. Although I have a good feel for this area in general, I did not carefully check the paper's details, e.g., the math, experimental design, or novelty.

---

> ### Author Rebuttal · Authors · 2023-08-25
>
> Thank you very much for your appreciation of our paper. Your comments, suggestions and questions are also very helpful and insightful, and we have provided the following responses to address them.
>
> **Response to Reject Reason1**:
>
> As an example, we compared the training and inference time between the Supervised baseline and the ADOROR method on the Persona-Zhao dataset in Table 1. Our machine configuration is as follows: CPU: Intel(R) Xeon(R) Gold 6242R. RAM: 32G. GPU: NVIDIA RTX 3060TI. The concurrent number of requests to the OpenAI LLM API is set to 20.
>
> Furthermore, we would like to emphasize that the time consumption of the ADOROR method mainly depends on the requests to the LLM API, which in turn depends on the speed of the API provided by OpenAI and the maximum concurrent number supported by your current environment. To speed up the training process, we have implemented caching for certain repetitive intermediate results, such as sentence embeddings and the results of requests with identical contexts.
>
> |                     | Training                                    | Inference                 |
> | ------------------- | ------------------------------------------- | ------------------------- |
> | Ruber-Roberta-Large  (we adopt a fine-tuning setup similar to [4]) | 20 minutes / 900 samples (1-fold, epoch=10) | 10 seconds  / 100 samples |
> | ADOROR              | 22 minutes / 900 samples (1-fold)           | 40 seconds / 100 samples  |
> Table1. Comparison of training  and inference speed on Persona-Zhao dataset.
>
> **Response to Reject Reason2**:
>
> Thank you for your suggestion. We will definitely consider incorporating these methods (such as BARTscore) in future versions, as long as the space of layout permits.
>
> **Response to Question1**:
>
> Yes, you are right. The phrase "from the same distribution" can be understood as referring to a batch of dialogue data generated in a similar manner (e.g., from self-chats of the same model, dialogues recorded by the same batch of annotators and machines). We chose to study the generalization of model scores under the same distribution rather than different distributions because our method relies on the in-context learning (ICL) ability of LLM, and the effectiveness of ICL is directly proportional to the similarity between the demonstration and query [1]. Compared to the Out-Of-Distribution (OOD) data, it is obviously easier to find demonstrations similar to the query within the same distribution. In extreme cases, such as when the two distributions are very different (such as gaming and medical scenarios), it is questionable whether ICL has transferability, so we first chose a pure same distribution evaluation scheme in our experiments. (Same-distribution evaluation has also been applied to other related work of dialogue evaluation, including supervised [3], semi-supervised [4], and self-supervised [5] ones.)
>
> Thank you very much for correcting our notation. We will make the changes in the future versions.
>
> **Response to Question2**:
>
> Yes, we set the decoding temperature to 0.0 and top-p to 1.0, expecting the model to perform greedy decoding. However, in reality, we found that temperature=0.0 is not always equivalent to greedy decoding. According to the official explanation from OpenAI, the model's output can be affected by the non determinism in GPU calculation [2], which can result in different outputs for the same context in rare cases. To ensure maximum reproducibility of our results, we cached all requests locally, meaning that the same context was only requested once.
>
> **References**
>
> [1] Liu, J., Shen, D., Zhang, Y., Dolan, W.B., Carin, L. and Chen, W., 2022, May. What Makes Good In-Context Examples for GPT-3?. In *Proceedings of Deep Learning Inside Out (DeeLIO 2022): The 3rd Workshop on Knowledge Extraction and Integration for Deep Learning Architectures* (pp. 100-114).
>
> [2] https://community.openai.com/t/a-question-on-determinism/8185
>
> [3] Lowe, R., Noseworthy, M., Serban, I.V., Angelard-Gontier, N., Bengio, Y. and Pineau, J., 2017, July. Towards an Automatic Turing Test: Learning to Evaluate Dialogue Responses. In *Proceedings of the 55th Annual Meeting of the Association for Computational Linguistics (Volume 1: Long Papers)* (pp. 1116-1126).
>
> [4] Zhao, T., Lala, D. and Kawahara, T., 2020, July. Designing Precise and Robust Dialogue Response Evaluators. In *Proceedings of the 58th Annual Meeting of the Association for Computational Linguistics* (pp. 26-33).
>
> [5] Zhang, C., Lee, G., D’Haro, L.F. and Li, H., 2021. D-score: Holistic dialogue evaluation without reference. *IEEE/ACM Transactions on Audio, Speech, and Language Processing*, *29*, pp.2502-2516.

---

### Official Review · Reviewer_JKu9 · 2023-08-06

**Typos Grammar Style And Presentation Improvements:** page 6 second column last paragraph t…
**Soundness:** 4

**Excitement:**

4: Strong: This paper deepens the understanding of some phenomenon or lowers the barriers to an existing research direction.

**Paper Topic And Main Contributions:**

Pre-trained dialog evaluation metrics do not scale well to different dimensions and possibilities available today with use of LLMs.  This paper proposes design and development of a dynamic dialog evaluation method to enable efficient adaptation to novel evaluation dimensions or unique scenarios. The paper utilizes an ICL-enhanced prompt generation method for training the black-box LLM and at inference, the LLM is queried multiple times alongwith a subgraph of diverse demonstrations to generate responses. Dialog evaluation method is compared against multiple existing metrics and shows promising results.

**Questions For The Authors:**

- it's hard to follow how many inversion prompts are being used in the overall training dataset.
- It is not clear how the method will perform on out of domain conversations.

**Reasons To Accept:**

- This paper presents a novel method for dialog evaluation using automatic prompt selection, demonstration selection and using the LLM multiple times with various demonstrations for dialog scoring. The method performs surprisingly better than many existing dialog evaluation benchmarks and looks very promising.
- Overall the method seems to correlate well with human scoring using the gpt3.5 human aligned models to help with automatic scoring.

**Reasons To Reject:**

- There's over reliance on the LLM to trust the automated scoring with the knowledge that LLMs have their complex biases and sensitivity to prompts (and order).
- It is not clear how the method will perform on long conversations (the dialog datasets used for prompt and demonstration selection seem to contain short conversations)
- The paper can be simplified in writing - the abstract is too long and does not convey the findings well.
- Fig. 1 can also be drawn better to show the processing pipeline (prompt generation and manual check, demonstration selection with ground truth scores, and automatic scoring alongwith showing where model training is being used to optimize the selection modules.

**Reproducibility:**

4: Could mostly reproduce the results, but there may be some variation because of sample variance or minor variations in their interpretation of the protocol or method.

**Reviewer Confidence:**

4: Quite sure. I tried to check the important points carefully. It's unlikely, though conceivable, that I missed something that should affect my ratings.

---

> ### Author Rebuttal · Authors · 2023-08-25
>
> We would like to express our sincere gratitude for your thorough and insightful review comments and questions. In response to the shortcomings and related issues you have pointed out, we have provided detailed explanations and clarifications.
>
> **Response to Reject Reason1**:
>
> Yes, we agree with this view, but we believe that there are several paths to improve the reliability of LLM and reduce bias:
>
> 1. Improve the diversity [2] and quality [1] of human-scored samples to better align with human scoring when conducting in-context learning.
> 2. Increase the number of human-scored samples, especially for sensitive data. With a sufficient amount of data, our framework can automatically find prompts that can reduce LLM bias.
> 3. Perform RLHF or instruction tuning before using LLM (of course, this path is cost-intensive, and the first two paths are more practical).
>
> **Response to Reject Reason2**:
>
> Yes, this is one of the limitations of our work because the currently available open-source dialogue datasets with good annotation consistency generally contain short dialogues. We are actively working on building a dialogue scoring dataset with better dialogue quality and longer dialogues.
>
> **Response to Reject Reason3**:
>
> Thank you very much for your suggestion. We will simplify the content of the abstract and adjust its structure to highlight the key points of our work.
>
> **Response to Reject Reason4**:
>
> Thank you for your suggestion. We will make adjustments to Figure 1 to better illustrate the important steps in our pipeline.
>
> **Response to Question1**:
>
> We apologize for not explicitly mentioning the number of inversion prompts in our paper. We will update this information in future versions. For each evaluation dimension in the dataset, we tune and keep one best inversion prompt. In practice, we further simplify the process by designing two inversion prompt templates for turn evaluation and dialogue evaluation, respectively, in which different {evaluation dimension} can be filled in (e.g., …Your objective is to construct a task [prompt] that can generate an {evaluation dimension} score…). Therefore, it is very convenient to use, as we only need to adjust two templates manually.
>
> **Response to Question2**:
>
> We believe that if the human demonstrations come entirely from others domains, it will significantly increase the difficulty of LLM in performing in-context learning. It is unknown whether our method will still be effective as our method highly relies on the in-context learning capability of the LLM. Some works have shown that the effectiveness of in-context learning is directly proportional to the similarity between the demonstration and the query [3]. In extreme cases, such as when two domains are very different (such as gaming and medical scenarios), it is questionable whether the LLM's ICL capability can still transfer well to different domains, while it is obviously much easier to find demonstrations similar to the query from the in-domain data. Therefore our initial work chose a relatively pure in-domain evaluation scheme. (In-domain evaluation has also been applied to other related work of dialogue evaluation, including supervised [4], semi-supervised [5], and self-supervised [6] ones.)
>
> Nevertheless, we also believe that evaluating OOD performance is important. One of the core contributions of our work is to replace human annotation with AI, but one of the prerequisites is the need for a batch of seed data manually generated by humans. If there is a need to frequently annotate dialogues in new domains, then it would be very valuable if the model can generalize to new domains. When the method has good OOD generalization performance, we can reuse the previously annotated data (from other domains), which can further reduce labor costs.
>
> **References**
>
> [1] Chen, J., Chen, L. and Zhou, T., 2023. It Takes One to Tango but More Make Trouble? In-Context Training with Different Number of Demonstrations. *arXiv preprint arXiv:2303.08119*.
>
> [2] Levy, I., Bogin, B. and Berant, J., 2022. Diverse demonstrations improve in-context compositional generalization. *arXiv preprint arXiv:2212.06800*.
>
> [3] Liu, J., Shen, D., Zhang, Y., Dolan, W.B., Carin, L. and Chen, W., 2022, May. What Makes Good In-Context Examples for GPT-3?. In *Proceedings of Deep Learning Inside Out (DeeLIO 2022): The 3rd Workshop on Knowledge Extraction and Integration for Deep Learning Architectures* (pp. 100-114).
>
> [4] Lowe, R., Noseworthy, M., Serban, I.V., Angelard-Gontier, N., Bengio, Y. and Pineau, J., 2017, July. Towards an Automatic Turing Test: Learning to Evaluate Dialogue Responses. In *Proceedings of the 55th Annual Meeting of the Association for Computational Linguistics (Volume 1: Long Papers)* (pp. 1116-1126).
>
> [5] Zhao, T., Lala, D. and Kawahara, T., 2020, July. Designing Precise and Robust Dialogue Response Evaluators. In *Proceedings of the 58th Annual Meeting of the Association for Computational Linguistics* (pp. 26-33).
>
> [6] Zhang, C., Lee, G., D’Haro, L.F. and Li, H., 2021. D-score: Holistic dialogue evaluation without reference. *IEEE/ACM Transactions on Audio, Speech, and Language Processing*, *29*, pp.2502-2516.

---

### Official Review · Reviewer_r8LW · 2023-08-11

**Soundness:** 4

**Excitement:**

4: Strong: This paper deepens the understanding of some phenomenon or lowers the barriers to an existing research direction.

**Paper Topic And Main Contributions:**

This paper proposes ADOROR, a prompting-based method for scoring dialogue systems w.r.t. flexible evaluation dimensions leveraging GPT-3.5-turbo. Experiments on five dataset demonstrate ADOROR outperforms various previous scoring methods.

**Questions For The Authors:**

Line 428 mentions the general cost of GPT-3.5-turbo, is there any estimation on the total cost (and time spent) of evaluating on the datasets? Just tracing from the writing it is a bit hard to get a feeling of the cost of ADOROR.

**Reasons To Accept:**

- Study on prompting-based dialogue evaluation methods could benefit the community (e.g. this class of methods could be beneficial when it is expensive/hard to obtain a sufficient amount of human annotations for training evaluation models from scratch)
- The proposed approach has good performance compared to various baselines, supported with coherent evaluations with clear experimental details.


**Reasons To Reject:**

I agree with the authors that experimenting with large language models is relatively expensive, however since ADOROR is a *model-agnostic* method I still feel it's a bit jumpy to draw conclusions from evaluating solely on GPT-3.5-turbo (for example, how does RLHF/instruction tuning affect results? what about the quality of the backbone LLM? etc..).

While running a model for all experiments are expensive, it could be fairly helpful if the author could consider reporting a smaller-scale sanity check (e.g. on 1 dataset, etc..) with another model.

**Reproducibility:**

5: Could easily reproduce the results.

**Reviewer Confidence:**

4: Quite sure. I tried to check the important points carefully. It's unlikely, though conceivable, that I missed something that should affect my ratings.

---

> ### Author Rebuttal · Authors · 2023-08-25
>
> Thank you very much for your recognition of our work. We also believe that one of the main practical values of our work is to some extent alleviate the problem of expensive manual annotation in dialogue evaluation (the paradigm can be transformed from manual massive annotation to manual fine-grained annotation + AI automated annotation). Regarding the shortcomings and issues you mentioned, we have provided supplementary explanations and answers in the following text.
>
> **Response to Reject Reasons**:
>
> We conducted additional experiments on the PersonaZhao dataset using GPT-4 and compared the results with GPT3.5-turbo as follows:
>
> |                                      | Spearman correlation coefficient (Average over 5-fold cross validation) |
> | ------------------------------------ | ------------------------------------------------------------ |
> | Auto-P-Selected (5.s, gpt-4)         | 0.784                                                        |
> | ADOROR (gpt-4)                       | 0.817                                                        |
> | Auto-P-Selected (5.s, gpt-3.5-turbo) | 0.680                                                        |
> | ADOROR (gpt-3.5-turbo)               | 0.711                                                        |
> Table1:  Results on Persona-Zhao dataset
>
> From Table 1, it can be observed that our proposed core modules, the prompt/demo selector and the construction of optimal subgraph, still improve the scoring performance on a new LLM—GPT4. We conduct 5-fold cross-validation and find that the Spearman coefficient of **ADOROR** (GPT-4) are consistently higher than the baseline (**Auto-P-Selected**) on all test sets. This experimental result preliminarily demonstrates that our method has good LLM adaptability.
>
> Additionally, we also attempted to use other large language models, such as LLAMA-2 [1] and Claude-2-100k [2], but preliminary experiments showed that LLAMA-2 [1] could not effectively follow the correct score format as shown in the demonstrations, leading to difficulties in post-processing parsing. Claude-2 also had output format issues, and tended to output very low scores regardless of context variations, so we did not further experiment with these models.
>
> **Response to Questions**:
>
> We can take the Persona-Zhao dataset as an example. The Persona-Zhao dataset contains 900 dialogues, with an average of 5 rounds per dialogue, and each round is relatively short. The cost of reproducing all the results (few shot k = 5, LLM = gpt-3.5-turbo) in the paper is approximately 160 USD, with 40% used for generating and selecting the most suitable prompts, 45% used for generating the dataset $\mathcal{O}$, and the remaining 8% used for test set inference. The cost of the first step, tuning the inversion prompt, can be ignored. Since we use 5-fold cross-validation, **the cost divided by 5 is roughly the cost in the real scenario (32 USD)**, which we consider reasonable. For inference, suppose the number of tokens per dialogue is between 300 and 800, if we set the size of the optimal graph $n_{\mathcal{G}^{'}}$ to 5, the cost of inferring once (**ADOROR** annotating one response with gpt3.5-turbo) is approximately between **0.003-0.008 USD**. Compared to manual annotation, the cost of labeling can be **reduced by a factor of 20** — for the current task, the cost of manual annotation using Amazon Mechanical Turk is around 0.06 USD per response [3].
>
> |                     | Training                                    | Inference                 |
> | ------------------- | ------------------------------------------- | ------------------------- |
> | Ruber-Roberta-Large (we adopt a fine-tuning setup similar to [4]) | 20 minutes / 900 samples (1-fold, epoch=10) | 10 seconds  / 100 samples |
> | ADOROR              | 22 minutes / 900 samples (1-fold)           | 40 seconds / 100 samples  |
> Table2. Comparison of training  and inference speed on Persona-Zhao dataset.
>
> Also taking the Persona-Zhao dataset as an example, we can see the training and inference time of ADOROR in Table 2. (Our machine configuration is as follows: CPU: Intel(R) Xeon(R) Gold 6242R. RAM: 32G. GPU: NVIDIA RTX 3060TI. The concurrent number of requests to the OpenAI LLM API is set to 20.)
>
> **References**
>
> [1] Touvron, H., Martin, L., Stone, K., Albert, P., Almahairi, A., Babaei, Y., Bashlykov, N., Batra, S., Bhargava, P., Bhosale, S. and Bikel, D., 2023. Llama 2: Open foundation and fine-tuned chat models. *arXiv preprint arXiv:2307.09288*.
>
> [2] Wu, S., Koo, M., Blum, L., Black, A., Kao, L., Scalzo, F. and Kurtz, I., 2023. A Comparative Study of Open-Source Large Language Models, GPT-4 and Claude 2: Multiple-Choice Test Taking in Nephrology. *arXiv preprint arXiv:2308.04709*.
>
> [3] https://aws.amazon.com/sagemaker/data-labeling/pricing/
>
> [4] Zhao, T., Lala, D. and Kawahara, T., 2020, July. Designing Precise and Robust Dialogue Response Evaluators. In *Proceedings of the 58th Annual Meeting of the Association for Computational Linguistics* (pp. 26-33).

---

### Official Review · Reviewer_Hsj3 · 2023-08-12

**Soundness:** 4

**Excitement:**

4: Strong: This paper deepens the understanding of some phenomenon or lowers the barriers to an existing research direction.

**Paper Topic And Main Contributions:**

The authors propose method called ADOROR to auto evaluate data across various dimensions by leveraging power of in context learning of Large Language Models. They first do prompt generation by essentially asking it to LLM. Then evaluate the generated prompts & various sampled in-context examples in two-tower fashion to get value function for effective prompt & demonstration selection. Authors, then formulate this task of selecting demonstrations and prompt as optimal sub-graph problem. As the problem NP-Hard in nature, They do greedily select most value among the set of demonstrations at every iteration.

**Reasons To Accept:**

# Strengths

- Technically **well written** paper. The experiments & ablations are thoughtful and well documented.

- Baselines like Ruber, Auto-P, Human-P (with BM 25) are comprehensive and widely used by community for these setups.

- Method has been tested on wide range of datasets (FED (various dimensions), Topical-USR, Persona-USR, DailyDialog-Zhao).

- Two Tower setup enables faster interference for large number of combinations of demonstrations.

**Reasons To Reject:**

# Weakness

- While it is interesting that the context demonstrations & prompt selection (two towers) are important, The results in terms of performance of these formulation is not as significantly better as expected. It questions many of assumptions in this formulation in the first place.

- Sensitivity analysis for few-shot k by proposed method ADOROR vs  Auto-P-selected baseline in various datasets, the difference between spearman correlation scores is less than 0.05, which may not be statistically significant for proving one way or other.

- Authors report that **ADOROR without suboptimal graph performs on par with ADOROR  in one dimension of FED, DDZ, PZ, UTC, UPC**, which goes to question can the difference attributed to randomness rather than this formulation.

**Reproducibility:**

5: Could easily reproduce the results.

**Reviewer Confidence:**

3: Pretty sure, but there's a chance I missed something. Although I have a good feel for this area in general, I did not carefully check the paper's details, e.g., the math, experimental design, or novelty.

---

> ### Author Rebuttal · Authors · 2023-08-25
>
> We would like to express our special appreciation for your recognition of our model architecture design and experimental details. At the same time, your constructive criticism is also very valuable. It is true that our original paper does miss some information on significance tests and has few inappropriate wordings, as you pointed out in the three weaknesses. We will include the results of significance test in our future versions. Below, we have provided some additional statistics to address your concerns.
>
> **Response to Weakness1**:
>
> We apologize for not explicitly stating the results of significance test in the original paper. Here we use the **Wilcoxon Sign Rank test** [1] to determine if there are significant differences between any two methods, with the calculation function denoted as $f_w(\cdot, \cdot)$. We represent the results of any method as $R=({r_1, …, r_{|R|}})$, where $r_i$ represents the the value of Spearman’s correlation coefficient [3] between a method's ratings of and human ratings on a test set, and $|R|$ equals 5 times the number of evaluation dimensions of all concerned datasets (multiplied by 5 because we perform 5-fold cross-validation). Given two methods, A and B, their Wilcoxon Sign Rank test result can be represented as $f_w(R_A, R_B)$.
>
> Before listing the comparison results, we briefly review the meaning of each method abbreviation:
>
>  \- **Auto-P-Selected**: Generating and selecting the best prompt based on the inversion prompt, training set and the LLM. (**Module 1**, ours)
>
> \- **ADOROR w\o Optimal subgraph**:  **Module 1** (ours) + training demonstration/prompt selector (**Module 2**, ours)
>
> \- **ADOROR**: **Module 1** (ours) + **Module 2** (ours) + constructing optimal subgraph (**Module 3**, ours)
>
> We present in Table 1 the differences between **ADOROR w\o Optimal subgraph** and **Auto-P-Selected (5.s)**, with p-values obtained from the Wilcoxon Sign Rank test (it should be noted that these p-values are different from those mentioned in Table 1 Caption of the original paper, which correspond to Spearman's correlation coefficients). For the DDZ and PZ datasets, which have a small number of results (5), we omit their p-values [2].
>
> | Average Improvement (p-value)                                | FED                     | DDZ  | PZ    | UTC                    | UPC                   | AVG                    |
> | ------------------------------------------------------------ | ----------------------- | ---- | ----- | ---------------------- | --------------------- | ---------------------- |
> | **ADOROR w\o Optimal subgraph** vs. **Auto-P-Selected (5.s)** | 11.03% (p-value: 0.022) | 3.4% | 4.26% | 3.39% (p-value: 0.075) | 3.5% (p-value: 0.059) | 5.88% (p-value: 0.004) |
> Table.1 Performance on all datasets with few shot k = 5 (For AVG, we integrate the results of all datasets.).
>
> We can see from Table 1 that the introduction of demonstrations & prompt selector (**Module 2**) leads to an improvement averaging around 6%, and the improvement is statistically significant (p<0.05 when considering all datasets).
>
> **Response to Weakness2**:
>
> Similarly to the previous comparison, we perform Wilcoxon Sign Rank test to determine whether there was a significant difference between **ADOROR** and **Auto-P-Selected**. For each dataset and method, we integrate the results from few-shot-k=1 to few-shot-k=7 (k = 1, 3, 5, 7), resulting in 20 data points. As shown in Table 2, although the improvement of **ADOROR** over **Auto-P-Selected** (i.e., the introduction of **Module 2 & 3**) is not particularly large, it is stable and statistically significant. Based on the results, the wording (significantly better) used in Figure 10 caption of the original paper is not very inappropriate, and we will make corrections in future versions.
>
> | Average Improvement (p-value)             | DDZ                       | PZ                         |
> | ----------------------------------------- | ------------------------- | -------------------------- |
> | **ADOROR**  vs. **Auto-P-Selected** | 6.06% (p-value: 5.72e-06) | 5.74%  (p-value: 9.53e-06) |
> Table.2 Performance on DDZ and PZ datasets with few shot k = 1,3,5,7.
>
> **Response to Weakness3**:
>
> Using the same method, we can see from Table 3 (first row) that constructing the optimal subgraph (**Module 3**) steadily improve the performance (though the improvement is minimal on the PZ dataset). The results are statistically significant when considering data points from all datasets.
>
> In addition, our core contribution is the proposal of **Module 2** and **Module 3**. Although the improvement of each individual module is not particularly large, the improvement becomes more pronounced when we integrate **Module 2** and **Module 3** (see Table 3, second row).
>
> | Average Improvement (p-value)                   | FED                       | DDZ   | PZ     | UTC                    | UPC                    | AVG                         |
> | ----------------------------------------------- | ------------------------- | ----- | ------ | ---------------------- | ---------------------- | --------------------------- |
> | **ADOROR**  vs. **ADOROR w\o Optimal subgraph** | 10.56% (p-value: 0.009)   | 2.66% | 0.282% | 2.13% (p-value: 0.074) | 1.00% (p-value: 0.139) | 6.34% (p-value: 0.0002)     |
> | **ADOROR**  vs.  **Auto-P-Selected (5.s)**      | 22.72%  (p-value: 0.0001) | 6.23% | 4.5%   | 5.6% (p-value: 0.036)  | 4.7% (p-value: 0.028)  | 14.32% (p-value: 5.542e-07) |
> Table.3 Performance  on all datasets with few shot k = 5.
>
> **References**
>
> [1] Woolson, R.F., 2007. Wilcoxon signed‐rank test. *Wiley encyclopedia of clinical trials*, pp.1-3.
>
> [2] Dwivedi, A.K., Mallawaarachchi, I. and Alvarado, L.A., 2017. Analysis of small sample size studies using nonparametric bootstrap test with pooled resampling method.
>
> [3] Sedgwick, P., 2014. Spearman’s rank correlation coefficient. BMJ: British Medical Journal, 349, p.g7327.

---

### Meta-Review · Area_Chair_krUM · 2023-10-04

**Recommendation:** 5

**Metareview:**

Very useful paper on the customization of very-long prompts to improve in-context learning. The paper is principled, well-written and impactful.

---

### Decision · Program_Chairs · 2023-10-07

**Decision:**

Accept-Main

**Comment:**

Very useful paper on the customization of very-long prompts to improve in-context learning. The paper is principled, well-written and impactful.